# Visualizing trypanosomes in a vertebrate host reveals novel swimming behaviours, adaptations and attachment mechanisms

Éva Dóró[1†], Sem H Jacobs[1], Ffion R Hammond[1‡], Henk Schipper[2], Remco PM Pieters[2], Mark Carrington[3], Geert F Wiegertjes[1,4], Maria Forlenza[1]*

[1]Department of Animal Sciences, Cell Biology and Immunology Group, Wageningen University & Research, Wageningen, Netherlands; [2]Department of Animal Sciences, Experimental Zoology Group, Wageningen University & Research, Wageningen, Netherlands; [3]Department of Biochemistry, University of Cambridge, Cambridge, United Kingdom; [4]Department of Animal Sciences, Aquaculture and Fisheries Group, Wageningen University & Research, Wageningen, Netherlands

**Abstract** Trypanosomes are important disease agents of humans, livestock and cold-blooded species, including fish. The cellular morphology of trypanosomes is central to their motility, adaptation to the host's environments and pathogenesis. However, visualizing the behaviour of trypanosomes resident in a live vertebrate host has remained unexplored. In this study, we describe an infection model of zebrafish (*Danio rerio*) with *Trypanosoma carassii*. By combining high spatio-temporal resolution microscopy with the transparency of live zebrafish, we describe in detail the swimming behaviour of trypanosomes in blood and tissues of a vertebrate host. Besides the conventional tumbling and directional swimming, *T. carassii* can change direction through a 'whip-like' motion or by swimming backward. Further, the posterior end can act as an anchoring site in vivo. To our knowledge, this is the first report of a vertebrate infection model that allows detailed imaging of trypanosome swimming behaviour in vivo in a natural host environment.
DOI: https://doi.org/10.7554/eLife.48388.001

*For correspondence:
maria.forlenza@wur.nl

Present address: †Institute of Transdisciplinary Discoveries, Medical School, Faculty of Medicine, University of Pécs, Pécs, Hungary; ‡Department of Infection, Immunity and Cardiovascular Disease, University of Sheffield, Sheffield, United Kingdom

Competing interests: The authors declare that no competing interests exist.

## Introduction

*Trypanosoma* is a genus of flagellated protozoa, and most species proliferate in the blood and tissue fluids of their host. The best-known and most commonly studied trypanosomes are those that cause human diseases: Human African Trypanosomiasis caused by *Trypanosoma brucei* and American Trypanosomiasis caused by *T. cruzi*. However, trypanosomes infect animals from all vertebrate classes, including warm-blooded mammals and birds as well as cold-blooded amphibians, reptiles and fish (*Simpson et al., 2006*).

African trypanosomes, such as *Trypanosoma brucei*, are exclusively extracellular and are continuously exposed to the innate, humoral and cellular immune systems. *T. brucei* has become a textbook example of antigenic variation based on low frequency switches of the expressed variant surface glycoprotein (VSG) that dominates the cell surface and allows the population to escape the host adaptive immune system. In addition, individual cell survival is favoured by a process that cleans the cell surface of low levels of bound immunoglobulins through endocytosis and degradation (*Engstler et al., 2007*; *Forlenza et al., 2009*). Trypanosome motility is an integral part of this process as the hydrodynamic drag on antibody-VSG complexes, caused by the forward swimming motion of the cell, results in accumulation of the complexes at the posterior pole, close to the flagellar pocket, where endocytosis occurs (*Engstler et al., 2007*). In addition, motility is essential for

**eLife digest** Trypanosomes are one-celled parasites that cause the disease trypanosomiasis, which is also known as sleeping sickness. Trypanosomiasis is transmitted to humans and animals by a type of fly, known as tse-tse, which is commonly found in sub-Saharan Africa. A bite from the tse-tse fly transfers the trypanosome cells into the host's bloodstream, where they spread from the blood to the internal organs and brain. This leads to a long-term illness, which can sometimes result in a coma and eventually death.

Once in the blood trypanosomes move around using a structure similar to an underwater propeller called the flagellum. How the trypanosomes move and behave in the blood determines how the infection will progress. Until now it has only been possible to observe trypanosomes in plastic dishes or in blood drawn from infected patients. However, neither of these settings mimic the conditions of the bloodstream, and it is currently impossible to look inside human hosts to watch how trypanosomes move.

To overcome this hurdle, Doro et al. infected zebrafish with *Trypanosoma carassii*, a close relative of the sub-Saharan trypanosomes that specifically infects fish. Zebrafish are transparent when young, making it possible to observe the parasite in the blood and tissues of live fish using a microscope.

Doro et al. noticed that *Trypanosoma carassii* cells adapt to different environments in the host by using different swimming techniques. For example, in small capillaries trypanosomes were dragged along with the blood flow, whilst in larger vessels, when blood flow was slow or there were fewer red blood cells, trypanosomes actively swam against the current. The parasites were also able to change direction by using their flagella in a 'whip-like' motion. Lastly, it was discovered that *Trypanosoma carassii* could rapidly attach to blood vessel walls using one end of its cell body, even when blood flow was strong. This behaviour may help the parasites escape from the bloodstream into the surrounding tissues, making the infection more dangerous.

Studying how trypanosomes infect zebrafish at this high level of detail provides new insights into how these parasites move and behave inside a host. An important question that remains to be answered, is how exactly the trypanosomes leave the bloodstream. A better understanding of the whole infection process may hint at new ways of fighting these deadly infections in future.
DOI: https://doi.org/10.7554/eLife.48388.002

successful cell division, immune evasion and development in the host (*Broadhead et al., 2006*; *Griffiths et al., 2007*; *Ralston et al., 2006*; *Shimogawa et al., 2018*).

In vitro studies using African trypanosomes have focused on the characterization of qualitative and quantitative parameters of trypanosome morphology and motility and emphasized the ability of trypanosome species to adapt to the various environments of their mammalian hosts (*Alizadehrad et al., 2015*; *Bargul et al., 2016*; *Heddergott et al., 2012*; *Krüger and Engstler, 2015*). Thus far, detailed analysis of swimming behaviour, morphology and trypanosome-host cell interaction has been restricted to ex vivo, using conditions that best mimic the host environment, for example in blood taken from naïve or infected animals (*Bargul et al., 2016*; *Beattie and Gull, 1997*; *Engstler et al., 2007*; *Heddergott et al., 2012*; *Hemphill and Ross, 1995*; *Shimogawa et al., 2018*; *Skalický et al., 2017*; *Sunter and Gull, 2016*; *Wakid and Bates, 2004*). It has not been feasible to recreate complex in vivo microenvironments, which, in the case of the vertebrate host, includes the streaming of the blood, vessels with heterogeneous size and endothelium composition, as well as changes occurring during the course of an infection, for example from the onset of anaemia. It still remains a challenge to mimic in vitro the dynamic conditions of the crowded and fast-moving bloodstream in which trypanosomes live. Given the importance of motility for trypanosome survival, analysis of trypanosome swimming behaviour in vivo, in a vertebrate host environment, is important to fully understand trypanosome biology and pathogenesis.

Quantitative and qualitative approaches in vivo are hindered by the lack of transparency of mammalian vertebrate hosts. Nevertheless, using transgenic trypanosomes expressing a luciferase reporter protein it has been possible to monitor overall trypanosome distribution and to quantify total trypanosome load in mice (*Burrell-Saward et al., 2015*; *Capewell et al., 2016*; *Goyard et al., 2014*; *McLatchie et al., 2013*). Fluorescent trypanosomes have been used in vivo to visualize their

presence in mouse skin, a possible reservoir of trypanosomes in the late phase of infection (*Capewell et al., 2016*). Although the sensitivity of both systems allows for indirect visualization of trypanosome location or distribution, low spatio-temporal resolution and the lack of transparency of deep tissues limits the possibility for qualitative and quantitative analysis of trypanosome swimming behaviour in vivo. Finally, using a combination of multicolour light sheet fluorescence microscopy and high-speed fluorescence microscopy it has been possible to analyse the infection process and swimming behaviour of *T. brucei* ex vivo, in dissected tissues of the partially transparent tsetse fly vector (*Gibson and Peacock, 2019*; *Schuster et al., 2017*; *Wang and Belosevic, 1994*). Nevertheless, to date, no method is available to study with sufficient resolution trypanosome swimming behaviour in vivo in a vertebrate host environment.

In the current study, we have used *Trypanosoma carassii* infection of zebrafish, *Danio rerio*, to visualize trypanosome movement in the bloodstream and tissues of a vertebrate host. *T. carassii* infects a broad range of cyprinid fish (*Overath et al., 1998*), is transmitted by blood-sucking leeches (i.e., *Hemiclepsis marginata*) (*Lom and Dyková, 1992*), and lives extracellularly in the blood and tissue fluids of the fish (*Haag et al., 1998*; *Lom and Dyková, 1992*). *T. carassii* can establish a long-term infection characterized by polyclonal B cell activation and recurrent waves of parasitaemia, without expression of a uniform VSG-like surface coat (*Agüero et al., 2002*; *Joerink et al., 2007*; *Overath et al., 2001*; *Overath et al., 1999*). Phylogenetically, *T. carassii* belongs to the aquatic clade of Trypanosoma, a sister group that diverged from the Trypanosomatid lineage prior to the divergence of the Stercorarian and Salivarian trypanosomes that infect mammals (*Simpson et al., 2006*; *Stevens, 2008*). Morphologically, *T. carassii* isolated from fish has a single flagellum that emerges from the flagellar pocket at the cell posterior, is attached to the cell body and extends free at the anterior end, defining the anterior-posterior axis.

Zebrafish have been used as a model for developmental biology, as well as biomedical and neurobiology research (*Asnani and Peterson, 2014*; *Blackburn and Langenau, 2014*; *Cronan and Tobin, 2014*; *Goessling and North, 2014*; *Miyares et al., 2014*; *Nguyen-Chi et al., 2014*; *Phillips and Westerfield, 2014*; *Renshaw and Trede, 2012*; *Schartl, 2014*; *Torraca et al., 2014*; *Veinotte et al., 2014*; *Zon and Cagan, 2014*), and are a fresh water cyprinid fish closely related to many of the natural hosts of *T. carassii*. The great advantage of zebrafish is the transparency of the larvae, and of some juvenile and adult stages. Furthermore, mutant and transgenic lines including those marking blood vessels and relevant immune cell lineages are available (*Benard et al., 2015*; *Bertrand et al., 2010*; *Ellett et al., 2011*; *Langenau et al., 2004*; *Lawson and Weinstein, 2002*; *Page et al., 2013*; *Petrie-Hanson et al., 2009*; *Renshaw et al., 2006*).

The transparency of the zebrafish allows high-resolution, real-time imaging of trypanosome movement and host-parasite interaction in vivo, under conditions that exactly represent the natural environment during infection. In this study, by combining *T. carassii* infection of transparent zebrafish with high-speed microscopy, we provide the first description of trypanosome swimming behaviour in vivo in a vertebrate host. Our observations reveal that trypanosomes can rapidly adapt their swimming behaviour to the heterogeneous host environments. It was not possible to assign one preferred swimming behaviour to trypanosomes in either the bloodstream, tissues or other body fluids. Conditions such as the presence or absence of the blood flow or of red blood cells, the speed of the flow, the size of the blood vessel, the type of endothelium or epithelium lining the vessels or the tissues, as well as the compactness of the tissues, all influenced the swimming behaviour. Furthermore, we show that trypanosomes can change direction through backwards swimming and through a 'whip-like' motion. Finally, we were able to capture a novel mechanism through which trypanosomes attach to host cells or tissues. These observations greatly expand our knowledge on trypanosome swimming behaviour and show that trypanosomes can rapidly adapt to match the host environment.

# Materials and methods

## Key resources table

| Reagent type (species) or resource | Designation | Source or reference | Identifiers | Additional information |
|---|---|---|---|---|
| Gene (*Danio rerio*) | *elongation factor-1α (ef1a)* | NA | ZDB-GENE-990415–52 | template for primers for RQ-PCR analysis |
| Gene (*Trypanosoma carassii*) | *heat-shock protein-70 (hsp70)* | NA | GeneBank-FJ970030.1 | template for primers for RQ-PCR analysis |
| Strain, strain background (*Cyprinus carpio*) | Wild type common carp, R3xR8 strain | doi:10.1016/0044-8486(95)91961 T | | |
| Strain, strain background (*Danio rerio*) | Wild type zebrafish, AB strain | European Zebrafish Resource Center (EZRC) | https://www.ezrc.kit.edu/index.php | |
| Strain, strain background (*Danio rerio*) | casper strain | *White et al. (2008)* | | optically transparent |
| Strain, strain background (*Danio rerio*) | *Tg(fli:egfp)$^{y1}$ (casper)* | doi:10.1038/nrg888 | | optically transparent line, marking the vasculature with green fluorescent protein |
| Strain, strain background (*Trypanosoma carassii*) | TsCc-NEM strain | *Overath et al. (1998)* | | |

## Zebrafish lines and maintenance

Zebrafish were kept and handled according to the Zebrafish Book (zfin.org) and animal welfare regulations of The Netherlands. Adult zebrafish were reared at the aquatic research facility of Wageningen University and Research (Carus). Zebrafish embryos and larvae were raised in egg water (0.6 g/L sea salt, Sera Marin, Heinsberg, Germany) at 27°C with a 12:12 light-dark cycle. From 5 days post fertilization (dpf) until 14 dpf, larvae were fed once a day with Tetrahymena. Larvae older than 10 dpf were also fed daily with dry feed ZM-100 (zmsystem, UK). The zebrafish lines used in this study included: wild type AB, optically transparent casper lines (*White et al., 2008*), a transgenic *Tg(fli: egfp)$^{y1}$* line marking the vasculature (*Lawson and Weinstein, 2002*) or crosses thereof. All animals were handled in accordance with good animal practice as defined by the European Union guidelines for handling of laboratory animals (http://ec.europa.eu/environment/chemicals/lab_animals/home_en.htm). All animal work at Wageningen University was approved by the local experimental animal committee (DEC number 2014095).

## *Trypanosoma carassii* culture

*Trypanosoma carassii* (strain TsCc-NEM) was previously cloned and characterized (*Overath et al., 1998*) and maintained in our laboratory by syringe passage through common carp (*Cyprinus carpio*, R3xR8 strain; *Irnazarow, 1995*). To this end, adult common carp were infected by intraperitoneal injection of $1 \times 10^4$ *T. carassii*; approximately 3 weeks post-infection and before the parasitaemia reached $1 \times 10^6$/mL, carp were euthanized in 0.6 g/L tricaine methane sulfonate (TMS, Crescent Research Chemicals) and bled via the caudal vein using a final concentration of 10–20 Units heparin/mL of blood. Trypanosomes in blood were imaged immediately or blood was kept at 4°C overnight in siliconized tubes. The following morning, trypanosomes were enriched at the interface between the red blood cells and plasma, and this buffy coat was recovered and centrifuged at 600xg for 8 min at room temperature. Trypanosomes were resuspended in RPMI without L-glutamine and phenol red (Lonza, Verviers, Belgium). To separate trypanosomes from red blood cells, the suspension was loaded on top of a 100% Ficoll-Paque layer (GE Healthcare, Uppsala, Sweden) and centrifuged at room temperature for 20 min, at 800xg. Cells at the interphase were transferred to a new 50 mL tube and washed with RPMI. Trypanosomes were then resuspended at a density of $5 \times 10^5$–$1 \times 10^6$/mL and cultured in 75 or 165 cm$^2$ flasks in complete medium: 22.5% MEM with L-glutamine and phenol red (Gibco), 22.5% Leibovitz's L-15 medium without L-glutamine, with phenol red (Lonza,

Verviers, Belgium), 45% Hanks' Balanced Salt solution (HBSS) with phenol red (Lonza, Verviers, Belgium), 10% sterile water, completed with 10% pooled carp serum, 2% 200 mM L-glutamine (Fisher Scientific) and 1% penicillin-streptomycin solution (10.000:10.000, Fisher Scientific). Cultures were incubated at 27°C without $CO_2$. Trypanosomes were kept at a density below $5 \times 10^6$/mL and subcultured one to three times a week. Using this medium, *T. carassii* was kept in culture without losing infectivity for up to 2 months. For zebrafish infection, trypanosomes were cultured for 1 week and never longer than 3 weeks.

## Zebrafish infection with *Trypanosoma carassii* and morbidity signs

Cultured trypanosomes were centrifuged at 800xg for 5 min and resuspended in 2% polyvinylpyrrolidone (PVP, Sigma-Aldrich) prior to injection. PVP was used to increase the viscosity of the medium to ensure a homogenous trypanosome solution throughout the injection period. Trypanosome number in 1–2 nL drop size varied depending on the intended dose and was monitored at the beginning and after 50 injections using a Bürker counting chamber. Prior to injection, 5 dpf zebrafish larvae were anaesthetized with 0.017% ethyl 3-aminobenzoate methanesulfonate (MS-222, Tricaine, Sigma-Aldrich) in egg water, and injected intravenously. Experimental groups received either *T. carassii* resuspended in PVP solution or PVP solution alone as a negative control. Injected larvae were directly transferred into pre-warmed egg water and kept in the incubator at 27°C. Viability was monitored daily. During the course of the optimization of the infection model, we noticed that some fish displayed lethargic behaviour and no escape reflex to a pipette, such fish usually had a high parasitaemia leading to death. These clinical signs were therefore used to monitor morbidity and the progression of infection. When necessary, fish were removed from the experiment and euthanized with an overdose of anaesthetic (0.4% MS-222).

## Real-time quantitative PCR

At various time points after infection, three to six zebrafish larvae were sacrificed by an overdose of anaesthetic, pooled and transferred to RNA later (Ambion). Total RNA isolation was performed with the Qiagen RNeasy Micro Kit (QIAgen, Venlo, The Netherlands) according to the manufacturer's protocol. Next, 250–500 ng of total RNA was used as a template for cDNA synthesis using SuperScript III Reverse Transcriptase and random hexamers (Invitrogen, Carlsbad, CA, USA), following the manufacturer's instructions with an additional DNase step using DNase I Amplification Grade (Invitrogen, Carlsbad, CA, USA). cDNA was then diluted 25 times to serve as template for real-time quantitative PCR (RT-qPCR) using Rotor-Gene 6000 (Corbett Research, QIAgen), as previously described (*Forlenza et al., 2012*; *Forlenza et al., 2008*). Primers for zebrafish *elongation factor-1α* (*ef1a* Fw: 3'-CTGGAGGCCAGCTCAAACAT-5' and RV: 3'-ATCAAGAAGAGTAGTAGTACCG-5'; ZDB-GENE-990415–52) and *T. carassii heat-shock protein-70* (FW: 3'-CAGCCGGTGGAGCGCGT-5' 3'-AGTTCCTTGCCGCCGAAGA-5'; GeneBank-FJ970030.1) were obtained from Eurogentec (Liège, Belgium). Gene expression was normalized to the *ef1a* housekeeping gene and expressed relative to the time point PVP control.

## High-speed light microscopy, image and video analysis

For imaging of *T. carassii* swimming behaviour in vitro, a high-speed camera was mounted on an automated DM6b upright digital microscope (Leica Microsystems), controlled by Leica LASX software (version 3.4.2.) and equipped with 100x oil (NA 1.32), 40x (NA 0.85, DIC) and 20x (NA 0.8, DIC) short distance objectives (Leica Microsystems). For high-speed light microscopy a (12 bits) Photron APX-RS High Speed Camera (Photron, resolution (128 × 16) to (1024 × 1024) pixels), with Leica HC 1x Microscope C-mount Adapter was used, controlled by Photron FASTCAM Viewer (PFV) software (version 3.5.1). Images were acquired at a resolution of 900 × 900 or 768 × 880 pixels depending on the C-mount adapter. Trypanosomes were transferred to non-coated microscopic slides (Superforst, Thermo Scientific), covered with a 24 × 50 mm coverslip and imaged immediately and for no longer than 10 min.

Prior to imaging of *T. carassii* swimming behaviour in vivo, the high-speed camera was mounted on a DMi8 inverted digital microscope (Leica Microsystems), controlled by Leica LASX software (version 3.4.2.) and equipped with 40x(NA 0.6) and 20x(NA 0.4) long distance objectives (Leica Microsystems). For high-speed light microscopy a (8 bits) EoSens MC1362 High Speed Camera (Mikrotron

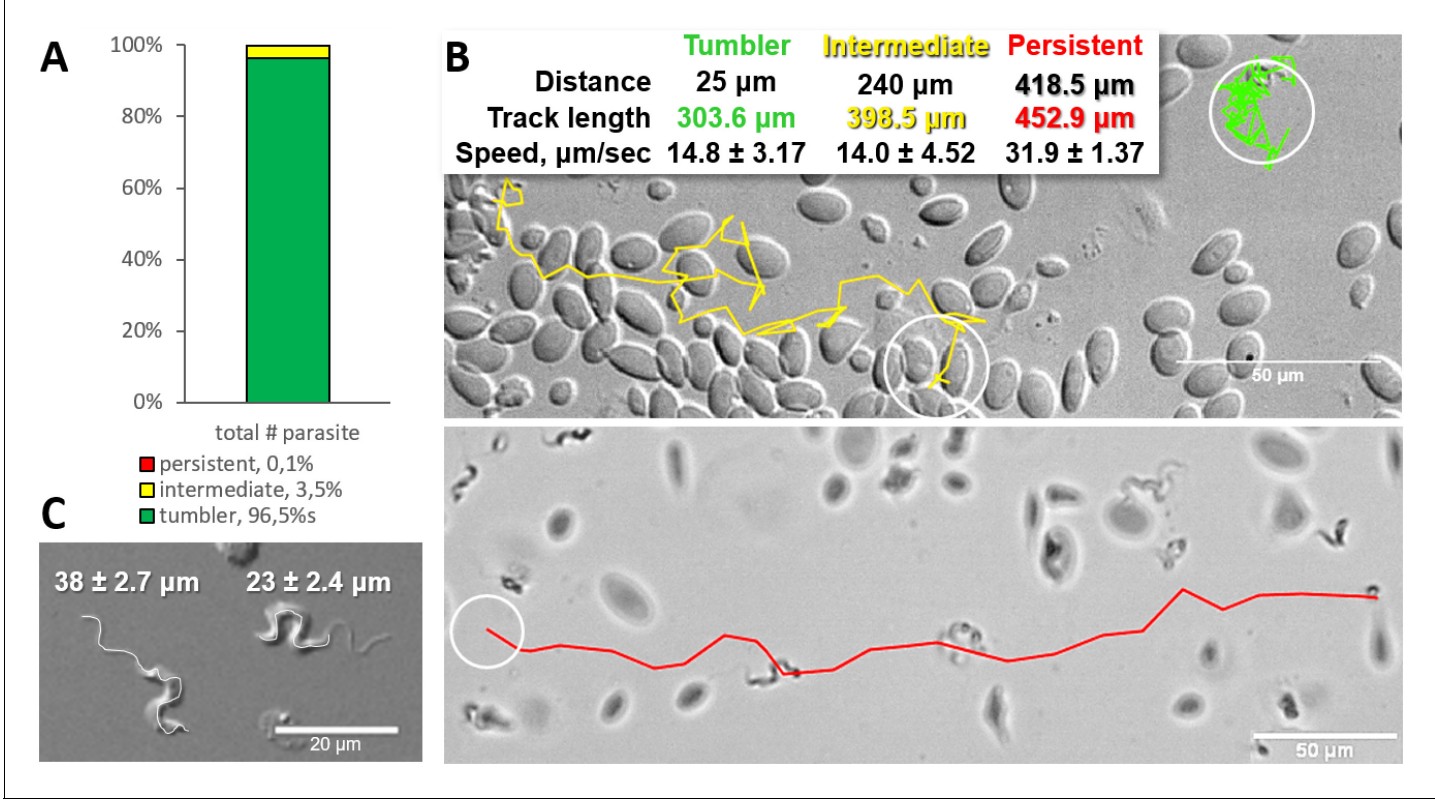

**Figure 1.** Majority of trypanosomes in freshly drawn blood are tumblers. Blood was freshly drawn from carp and *T. carassii* swimming behaviour analysed immediately using high-resolution microscopy at 240 frames per second (fps). (**A**) Relative percentage of tumblers, and intermediate or persistent swimmers (defined in the text) was calculated over a total number of 944 *T. carassii*, isolated from six different carp infections and imaged over 60 independent acquisitions. (**B**) Representative tracks of a tumbler (green), intermediate (yellow) and persistent (red) swimmer. The diameter of the circles (23 µm) indicates the average cell-body size of a trypanosome as also shown in (**C**). The inset table summarizes the straight-line distance covered by the trypanosome (between the first and last track point); the total track length, that is the path covered by the trypanosome in approximately 20 s of acquisition time, indicated in matching colours; and the average speed (µm/s) was calculated on a selection of the acquisitions used in (**A**). For tumblers, the displacement of the posterior end was used as tracking point. (**C**) Detailed image of two trypanosomes indicating the total body length including the flagellum (left) and the total cell-body length excluding the flagellum (right). Measurements were acquired on high-resolution images of at least 10 freshly isolated trypanosomes obtained from four independent infections, using more than 20 frames within the same acquisition. Quantification of trypanosome length, swimming speed and directionality was performed with ImageJ-Fijii using the MTrack plug-in. *Video 1* displays high-speed videos of the swimming behaviour of tumblers, intermediate and persistent swimmers in carp blood, or of trypanosomes in serum or culture medium.

DOI: https://doi.org/10.7554/eLife.48388.003

The following source data is available for figure 1:

**Source data 1.** Percentage of tumblers, intermediate and persistent swimmer.
DOI: https://doi.org/10.7554/eLife.48388.004

**Source data 2.** T. carassii speed in vitro.
DOI: https://doi.org/10.7554/eLife.48388.005

GmbH, resolution 1280 × 1024 pixels), with Leica HC 1x Microscope C-mount Camera Adapter was used, controlled by XCAP-Std software (version 3.8, EPIX inc). Images were acquired at a resolution of 900 × 900 or 640 × 640 pixels. Zebrafish larvae were anaesthetized with 0.017% MS-222 and embedded in UltraPure LMP Agarose (Invitrogen) on a microscope slide (1.4–1.6 mm) with a well depth of 0.5–0.8 mm (Electron Microscopy Sciences). Upon solidification of the agarose, the specimen was covered with three to four drops of egg water containing 0.017% MS-222, by a 24 × 50 mm coverslip and imaged immediately.

For all high-speed videos, image series were acquired at 240–500 frames per second (fps) and analysed using PFV software (version 3.2.8.2) or MiDAS Player v5.0.0.3 (Xcite, USA); 240–250 fps

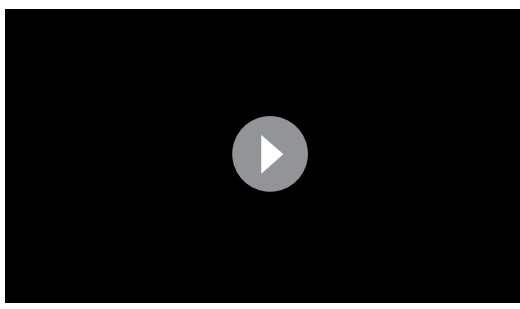

**Video 1.** Swimming behaviour of *T. carassii* in freshly drawn carp blood, or blood diluted with either carp serum or culture medium, showing representative tumblers, intermediate and persistent swimmers.
DOI: https://doi.org/10.7554/eLife.48388.006

were found optimal for imaging of trypanosome swimming behaviour in vitro, whereas 480–500 fps were used for in vivo imaging of infected zebrafish. Quantification of trypanosome length, swimming speed and directionality was performed with ImageJ-Fijii (version 1.51 n) using the MTrack plug-in. For livestream light microscopy (acquisitions at 20 fps) a DFC3000G camera (Leica Microsystems) was mounted on the DMi8 inverted digital microscope and controlled by the Leica LASX software. Images were acquired at a resolution of 720 × 576 or 1296 × 966 pixels.

For fluorescence microscopy of *Tg(fli1:egfp)^{y1}* casper lines, marking the zebrafish vasculature in green, the Zeiss LSM-510 confocal microscope, with a 20x long-distance objective was used with the following settings: laser excitation = 488 nm with 73% transmission; HFT filter = 488 nm; BP filter = 505–550; detection gain = 800; amplifier offset = −0.01; amplifier gain = 1.1; bright field channel was opened with Detection Gain = 130; frame size (pixels) = 2048×2048; pinhole = 300 (optical slice <28.3 μm, pinhole ø = 6.26 airy units). Videos were produced using CyberLink Power-Director 16.

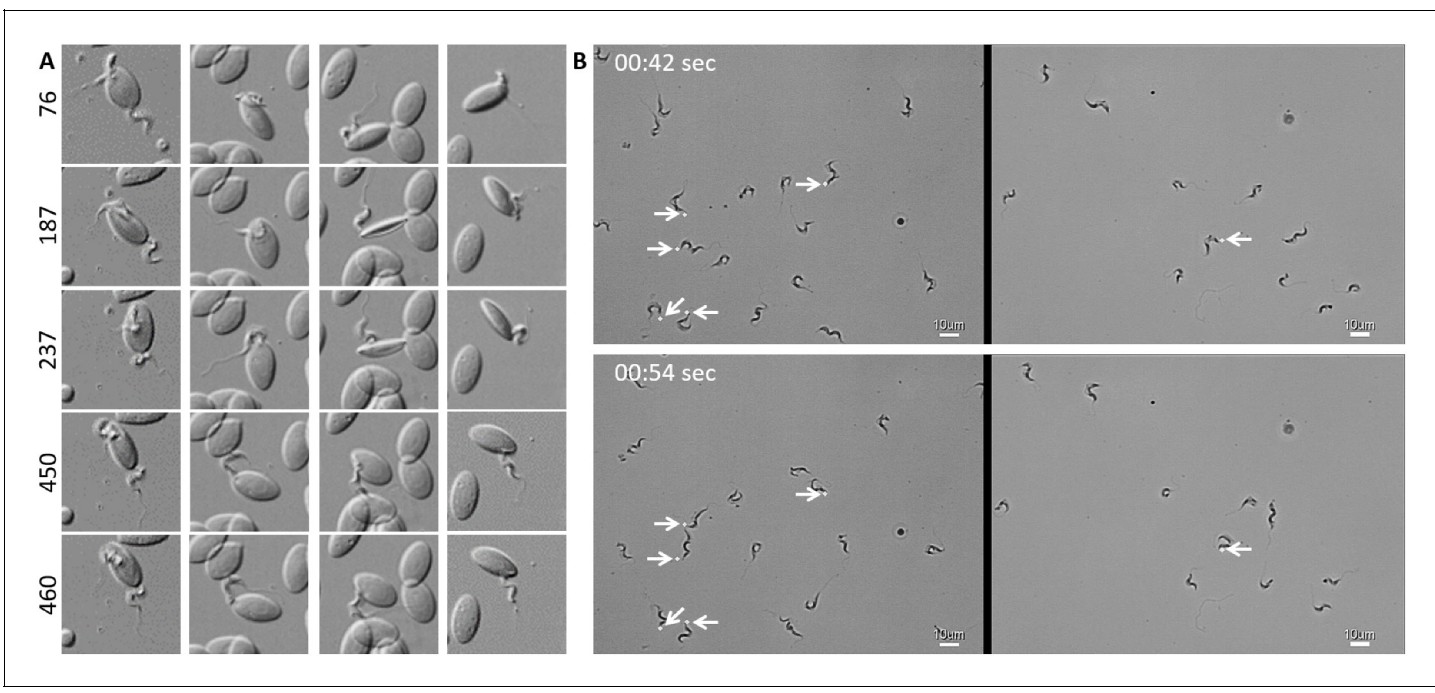

**Figure 2.** *T. carassii* attaches to cells or surfaces through its posterior end leaving the flagellum free to move. (**A**) Blood was freshly drawn from carp and trypanosomes' swimming behaviour immediately imaged using high-resolution microscopy at 240 frames fps. Images are frames (indicated by the numbers) of four different locations within the same field of view, selected from the corresponding *Video 2*. Note how the posterior end of the parasites is attached to the red blood cell and the flagellum is free to move. (**B**) Selected frames from *Video 2*, at the indicated time points, show how *T. carassii* can also adhere to glass surfaces through the posterior end (white arrow) leaving the body and flagellum free to move.
DOI: https://doi.org/10.7554/eLife.48388.007

The following source data is available for figure 2:

**Source data 1.** Susceptibility of zebrafish larvae to *T. carassii* infection and kinetics of parasitaemia.
DOI: https://doi.org/10.7554/eLife.48388.008

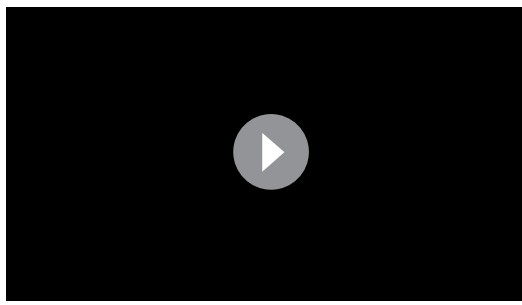

**Video 2.** *T. carassii* attaches to cells or surfaces through its posterior end leaving the flagellum free to move. Trypanosomes were also observed to swim backwards.
DOI: https://doi.org/10.7554/eLife.48388.009

## Results

### Characterization of *T. carassii* swimming behaviour in vitro

Previous in vitro studies reported on the heterogeneity in swimming behaviour of African trypanosomes, and on how this was dependent on the viscosity of the culture medium or host blood (*Bargul et al., 2016*; *Engstler et al., 2007*; *Heddergott et al., 2012*). To investigate the swimming behaviour of *T. carassii* in fish blood or culture medium, we used high spatio-temporal resolution microscopy. For the initial description of trypanosome swimming behaviour in vitro, we adopted the classification and quantification method described previously (*Bargul et al., 2016*): *persistent swimmers*, trypanosomes exhibiting a directional movement covering several hundreds of micrometres; *tumblers*, trypanosomes exhibiting a non-directional movement and travelling no further than their body length; and *intermediate swimmers,* trypanosomes alternating periods of directional and non-directional movement.

Analysis of the swimming behaviour of trypanosomes in blood of infected carp revealed that up to 96.4% of *T. carassii* could be classified as *tumblers* (*Figure 1A–B*, and *Video 1* (00:00 - 45:19 s)). The remaining trypanosomes (3.5%) behaved as *intermediate swimmers* (*Video 1*, 00:00 - 45:19 s), and only 0.1% could be classified as *persistent swimmers* (*Figure 1A–B* and *Video 1*, 45:19 - 58:03 s). Persistent swimmers showed an average speed of 32 µm/s and could cover a 'straight-line' distance of up to 418 µm in 20 s, whereas intermediate swimmers showed a lower average speed of 14 µm/s (*Figure 1B*). Although tumblers did not move any distance greater than their body length (~23 ± 2.4 µm, *Figure 1C*), they were still very mobile. Taking advantage of the spatio-temporal resolution of high-speed videography, the speed of displacement of the posterior end of tumblers was 14 µm/s, similar to that of intermediate swimmers (*Figure 1B*).

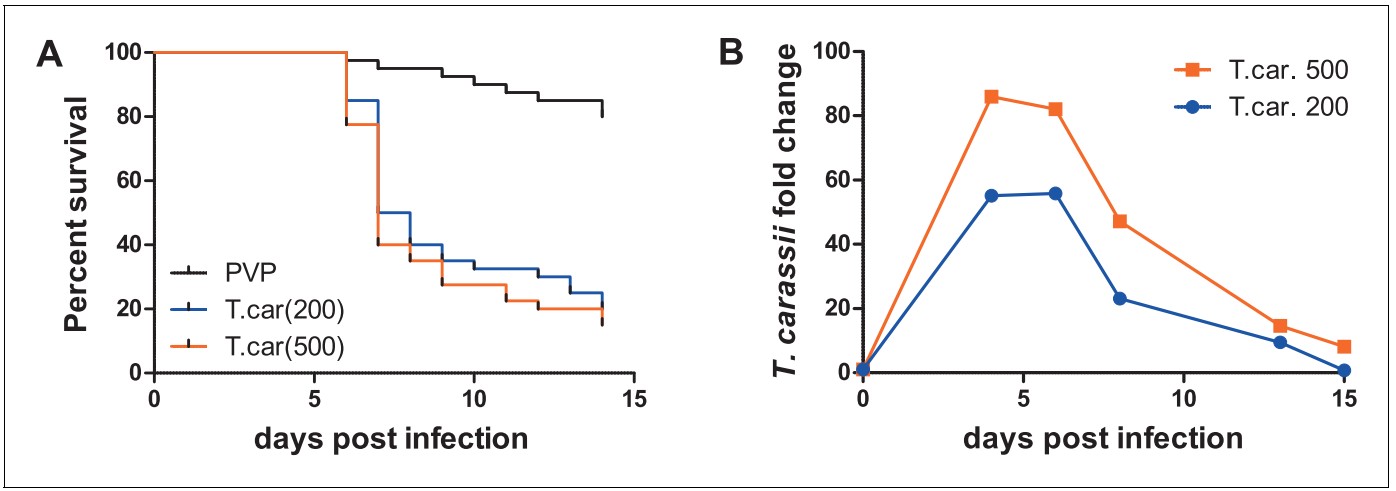

**Figure 3.** Zebrafish larvae are susceptible to *T. carassii* infection. Zebrafish larvae (5 dpf) were injected with the indicated number of *T. carassii* per fish. PVP was used as injection control. (**A**) Kinetics of zebrafish larval survival. Fish (n = 40/group) were observed daily for signs of infection and survival. (**B**) Kinetics of parasitaemia. Trypanosome levels were quantified by real-time quantitative-PCR using *T. carassii* hsp70-specific primers. RNA of five fish was pooled at each time point. Expression was normalized relative to the host house-keeping gene *elongation factor-1 alpha* and expressed relative to fish injected with the corresponding trypanosome dose at time point zero.
DOI: https://doi.org/10.7554/eLife.48388.010

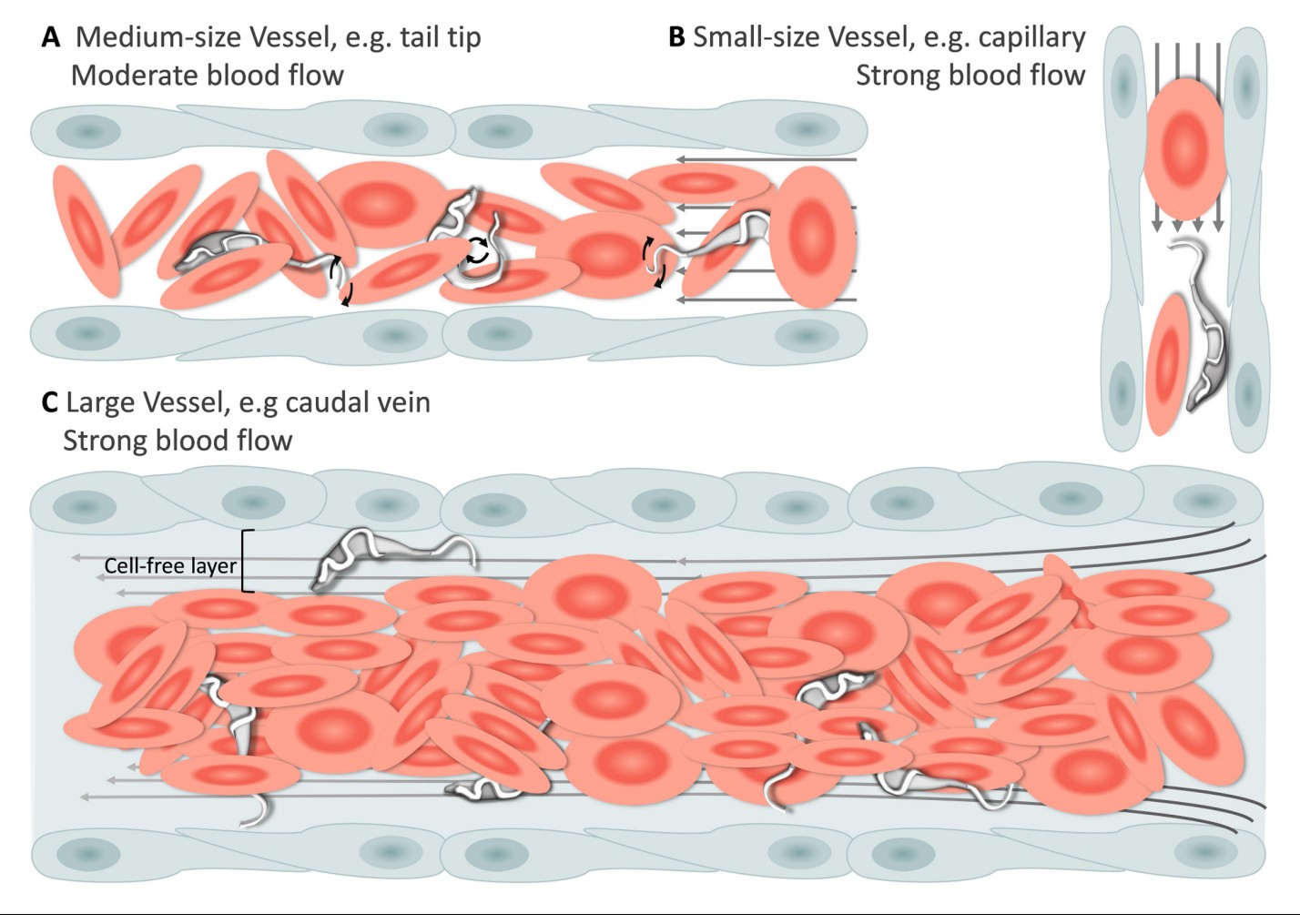

**Figure 4.** Schematic representation of *T. carassii* swimming behaviour in blood vessels of various sizes. In general, in vessels with an intact blood flow (grey arrows) and a normal number of red blood cells (RBC), trypanosomes are dragged passively by the flow along with RBC. Under these conditions, trypanosomes were never seen swimming directionally against the flow or faster than RBC. (**A**) In medium-sized blood vessels with moderate flow, while being dragged passively by the flow, trypanosomes can either curl or stretch their body, as well as occasionally propel their flagellum in the same or opposite direction to the blood flow. (**B**) In small-sized blood vessels of one-cell diameter, such as intersegmental capillaries (ISC), trypanosomes are pushed forward by the blood flow and by colliding RBC. (**C**) In large-sized blood vessels, the blood flow is very strong and the density of RBC high, making it more difficult to detect trypanosomes without the aid of high-speed microscopy. Only the occasional trypanosome that would slow down by bouncing against the vessel wall would be visible in the cell-free layer. *Video 3* contains high-speed videos showing details of the trypanosome movements schematically depicted above.

DOI: https://doi.org/10.7554/eLife.48388.011

Comparison of the movement of freshly isolated trypanosomes kept in either carp serum or culture medium revealed comparable swimming behaviour (*Video 1*, 00:58 s - 1:15 min). Culture in medium or serum for a period of up to 2 months did not alter morphology, the proportions of tumblers, intermediate or persistent swimmers (data not shown), suggesting that the swimming behaviour is an intrinsic property of trypanosomes in blood.

In freshly drawn blood from infected carp, trypanosomes were observed anchored to red blood cells or to leukocytes. Attachment always occurred through their posterior end leaving the flagellum free to move (*Figure 2* and *Video 2*, 00 - 28 s). Similarly, also when cultured, trypanosomes could attach to the flask or glass surface through their posterior end (*Video 2*, 28 - 55 s). Whether the site of attachment coincided with the cell membrane, flagellum base, or neck of the flagellar pocket was not readily clarified with the current image resolution. Finally, trypanosomes were also observed to alternate between forward and backward swimming (*Video 2*, 00:55 s - 1:15 min).

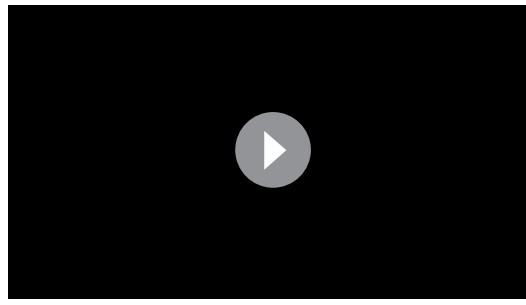

**Video 3.** Swimming behaviour of *T. carassii* in small, medium and large-sized vessels with an intact blood flow. Under these conditions, trypanosomes are dragged passively by the flow or pushed forward by colliding RBC.

DOI: https://doi.org/10.7554/eLife.48388.012

## Establishment of a *Trypanosoma carassii* infection model in zebrafish

To observe the swimming behaviour of *T. carassii* in a host, we developed an infection model in transparent zebrafish larvae. First, the susceptibility and kinetics of parasitaemia were determined. Infection of 5 day-post-fertilization (dpf) zebrafish resulted in an acute infection associated with low survival (*Figure 3A*) and high parasitaemia (*Figure 3B*), independent of the infection dose. This confirms that zebrafish, similarly to other cyprinid fish, are susceptible to *T. carassii* infection.

## In vivo, T. carassii rapidly adapts its swimming behaviour to the heterogenous blood environment

Having established that *T. carassii* can infect zebrafish, we took advantage of the transparency of zebrafish larvae to characterize trypanosome swimming behaviour in vivo. Zebrafish larvae were infected at 5 dpf with 200 *T. carassii* per fish and imaged using either live stream imaging (20 fps) or high spatio-temporal resolution microscopy (500 fps) at various time points after infection and in differently sized blood vessels. When parasitaemia was low, typically early during infection, trypanosomes were most readily detected in small to medium-sized blood vessels with reduced blood flow and a lower density of red blood cells, such as the tail tip loop or intersegmental capillaries (ISCs). In the tail tip loop, trypanosomes were dragged passively by the bloodstream along with red blood cells (*Figure 4A*, *Video 3*, 0 - 42 s) and were seen to either curl or stretch the cell body as well as occasionally propel their flagellum in the same or opposite direction to the blood flow (*Video 3*,

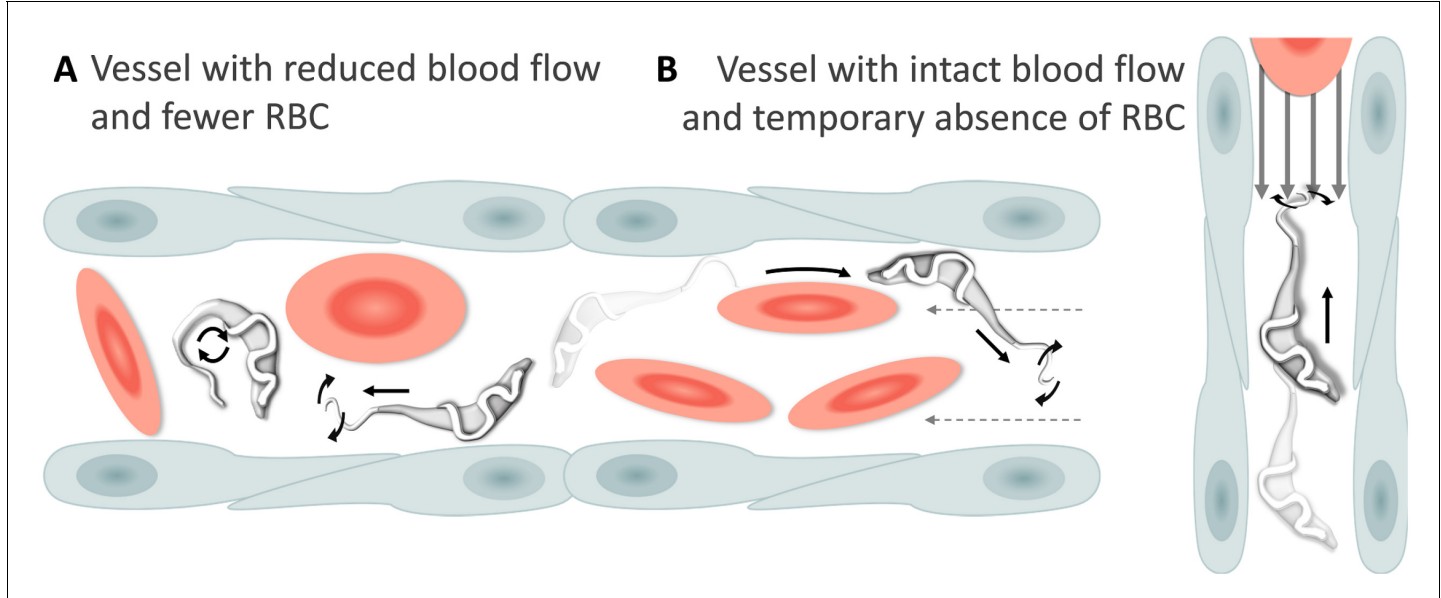

**Figure 5.** Schematic representation of *T. carassii* swimming behaviour in blood vessels with altered blood flow or red blood cell (RBC) number. (**A**) Blood vessels with a reduced blood flow (dashed arrows) and fewer RBC. (**B**) Blood vessels with an intact blood flow (grey arrows) but with a temporary absence of RBC. In both cases, trypanosomes can swim directionally (black straight arrows) by propelling the flagellum (distorted circular arrows) in the same or opposite direction to the flow, or can tumble (circular arrows). *Video 4* contains high-speed videos showing trypanosome movement in medium-sized blood vessels and in capillaries as schematically depicted.
DOI: https://doi.org/10.7554/eLife.48388.013

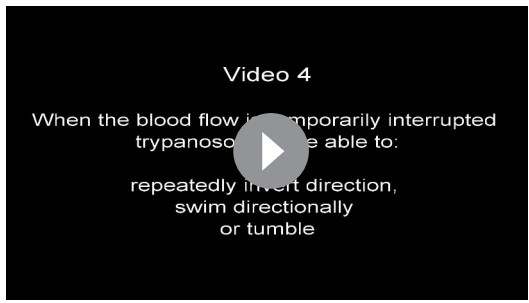

**Video 4.** Swimming behaviour of *T. carassii* in medium-sized blood vessels with interrupted blood flow, or in capillaries with intact blood flow and reduced number of red blood cells. Under these conditions, trypanosomes can repeatedly invert direction, swim directionally or tumble.
DOI: https://doi.org/10.7554/eLife.48388.014

00:42 s - 01:57 min), but were never seen swimming faster than the flow. ISCs are narrow, with a diameter equivalent to a single red blood cell. In ISCs, trypanosomes were elongated with their flagellum in the opposite direction to the blood flow (*Figure 4B*, *Video 3*, 01:57 - 02:49 min); the diameter of the vessel, the speed of the flow and the presence of colliding red blood cells within the ISC, force the trypanosomes to passively move forward in the direction of the flow.

In larger diameter vessels with a strong blood flow and higher density of red blood cells, such as the cardinal caudal vein (*Video 3*, 02:49 - 04:11 min) or artery (*Video 3*, 04:11 - 04:37 min), detection and description of swimming behaviour was greatly aided by the use of high spatio-temporal resolution microscopy. In these vessels as well, trypanosomes are dragged passively by the bloodstream, curling among the densely packed red blood cells. In real-time speed, only the occasional trypanosome was seen to slow down by rolling or bouncing against the vessel in the peripheral cell-free layer (*Figure 4C* and *Video 3*, 03:53 - 04:11 min). In general, the typical tumbling movement described in vitro was not observed in the fish, except for those locations where the blood flow was highly reduced or absent as in sharp turns of the tail tip (*Video 3*, 04:37 - 05:09 min),

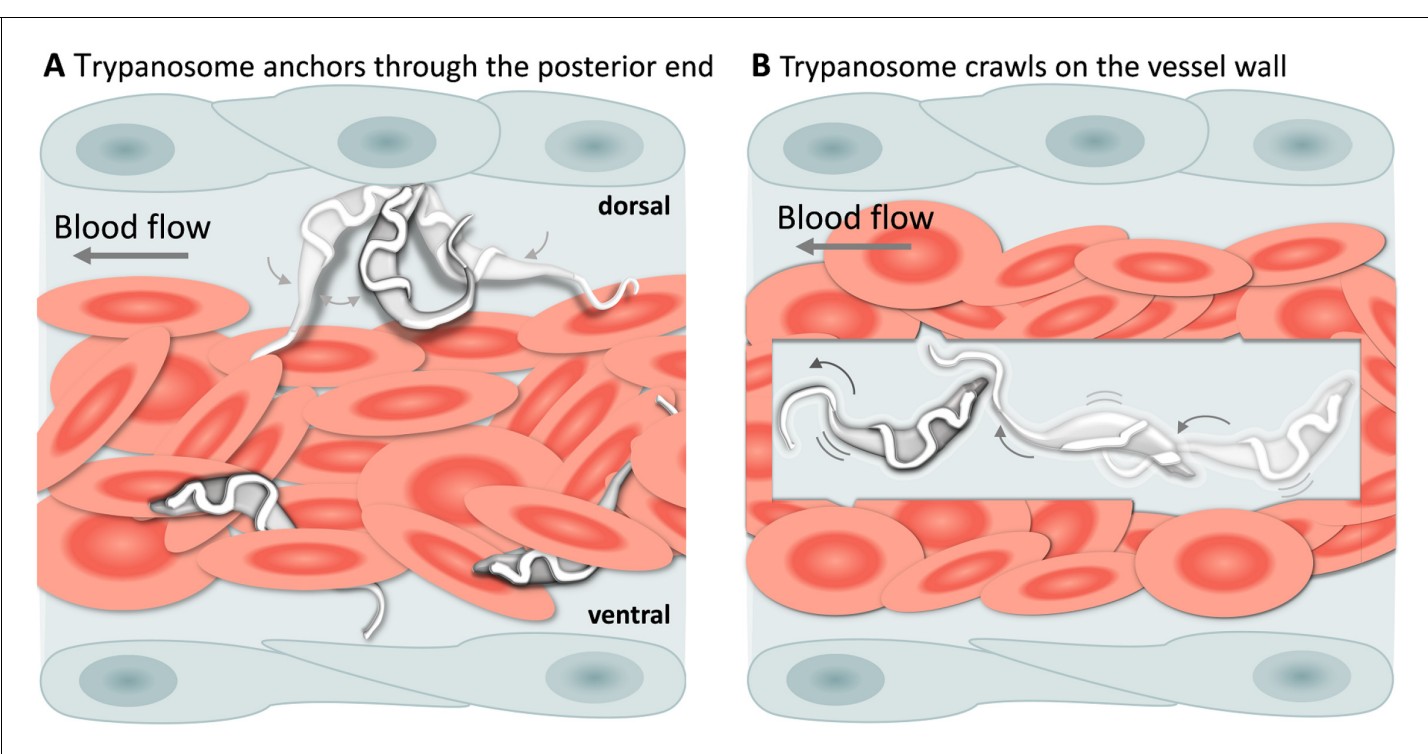

**Figure 6.** *T. carassii* attachment to the blood vessel wall. (A) *T. carassii* anchor themselves by their posterior end, leaving the cell body and the flagellum free to move. Anchoring occurred only to the dorsal luminal side of the caudal vein and was not observed in other blood vessels. (B) Trypanosomes crawl along the vessel wall of the caudal vein; the transparent square allows visualization of trypanosomes through the pack of red blood cells. Crawling occurred everywhere in the vein and involved the entire cell body. *Video 5* contains high-speed videos showing anchored and crawling *T. carassii* in the caudal vein as schematically depicted above.
DOI: https://doi.org/10.7554/eLife.48388.015

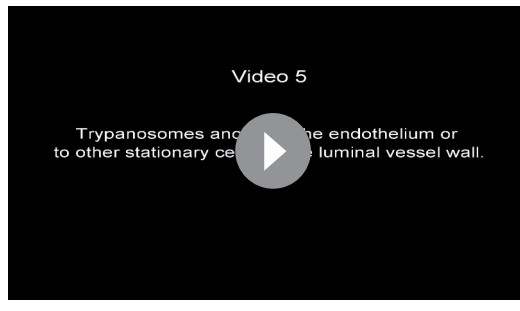

**Video 5.** *T. carassii* anchor to the dorsal luminal side of the caudal vein and crawl along the vessel wall of veins. *T. carassii* movement in fluids outside the blood vessels and in tissues.
DOI: https://doi.org/10.7554/eLife.48388.016

and in locations where leukocytes adhering to the endothelium would create a local disturbance of the flow rate allowing trypanosomes to tumble (*Video 3*, 05:09 - 05:24 min).

Physiological changes in the blood flow or red blood cells density associated with the infection, allowed visualization of additional swimming behaviours in blood vessels. For example, in cases where the blood flow was temporarily interrupted, trypanosomes were able to swim directionally, repeatedly invert direction, or tumble (*Figure 5A*, and *Video 4*, 0 - 01:23 min). In cases where the blood flow continued but red blood cells were occluded, trypanosomes were able to persistently swim against the blood flow (*Figure 5B*, and *Video 4*, 01:23 - 02:13 min). We did not attempt to distinguish, in vivo, between intermediate and persistent swimmers because physical factors such as the presence of red blood cells, the length of a capillary or the stability of the blood flow could all interfere with the directionality of their movement. Instead, all trypanosomes observed to swim directionally in the blood, independent of the distance covered before changing direction, were considered swimmers.

Altogether, we observed that *T. carassii* can adopt different swimming behaviours all greatly influenced by the blood flow, size of the blood vessel and presence of red blood cells. Most frequently, in blood vessels with an intact blood flow and high density of red blood cells, trypanosomes are dragged passively by the flow along with red blood cells, in a curling motion. Occasionally, when the blood flow or the number of red blood cells is reduced, trypanosomes can swim directionally and

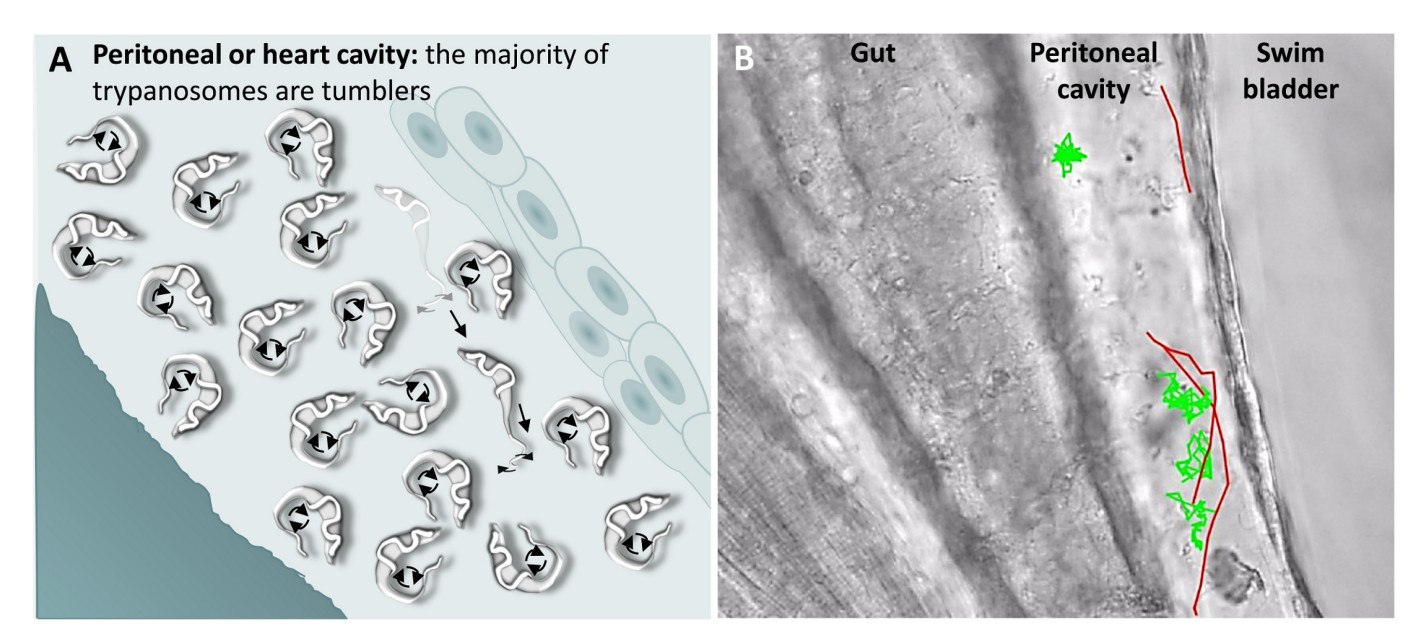

**Figure 7.** *T. carassii* swimming behaviour in tissue fluids outside blood. (A) Schematic representation of *T. carassii* swimming in the peritoneal or heart cavity, both environments without hydrodynamic flow and red blood cells; here, most of the trypanosomes are tumblers, only occasionally was a persistent swimmer observed. (B) Selected frame from *Video 6*, capturing trypanosomes in the peritoneal cavity. More than 100 trypanosomes are present but are not all in focus in the selected frame; the majority are tumblers. The tracks of four representative tumblers (green) and of the only three persistent swimmers (red) are shown. *Video 6* contains high-speed videos showing the location and swimming behaviours described above.
DOI: https://doi.org/10.7554/eLife.48388.017

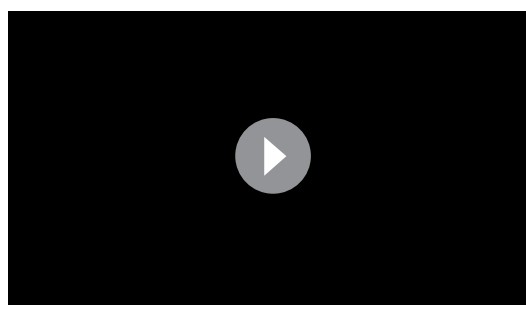

**Video 6.** Swimming behaviour of *T. carassii* extravasated from blood vessels into tissue fluids that lack blood flow and red blood cells. These include the peritoneal and heart cavities.
DOI: https://doi.org/10.7554/eLife.48388.018

persistently (swimmers) in blood vessels. This indicates that, at least in vivo, it is not possible to assign a specific swimming behaviour to trypanosomes in blood vessels; on the contrary, trypanosomes rapidly adapt their swimming behaviour to changes in microenvironmental conditions.

## Trypanosomes can attach in vivo by anchoring through their posterior end

In addition to being dragged passively within blood vessels, trypanosomes could often be seen attached to the endothelium on the dorsal luminal side of the cardinal caudal vein (referred to as caudal vein, *Figure 6A*). Remarkably, despite the strong blood flow, attachment (anchoring) could last for several seconds (*Video 5*, 0 - 01:11 min) and involved a small area of the posterior end of the trypanosome, leaving the cell body and the flagellum free to move (*Video 5*, 01:11 - 01:59 min). Although anchoring could be observed already at 1 dpi, it was more easily detected at later stages of the infection. Anchoring was not the only mode of attachment, trypanosomes were also seen crawling along the vessel wall involving the entire cell body (*Figure 6B*, and *Video 5*, 01:59 - 02:38 min). Within blood vessels, anchoring occurred exclusively at the dorsal side of the caudal vein, whereas crawling could occur anywhere in the vein. No attachment or crawling was observed in arteries or capillaries, independently of the speed of the blood flow.

Altogether, our observations show that *T. carassii* attaches to host cells through their posterior end, both in vitro (*Video 2*) and in vivo, leaving the flagellum free to move, and suggest that the posterior end acts as an anchoring site. Whether the exact anchoring site corresponds to the flagellum base or to the neck of the flagellar pocket and whether it may possibly favour extravasation, could not be confirmed under the current conditions and will be the focus of further investigation.

## *T. carassii* movement in fluids outside the blood vessels and in tissues

A characteristic of *T. carassii* infections is extravasation from blood vessels into surrounding tissues and tissue fluids (*Haag et al., 1998*; *Lom and Dyková, 1992*). In zebrafish, this was observed at 1 dpi and allowed us to investigate the swimming behaviour in tissue fluids other than blood, including those of the peritoneal and heart cavities. In these locations, we could thus evaluate the swimming behaviour of trypanosomes in the absence of red blood cells and of blood flow. In these environments, the swimming behaviour was similar to that observed in vitro (*Video 1* and *Figure 1*), where the majority of the trypanosomes were tumblers (*Figure 7* and *Video 6*, 0 – 42 s). In a field of view of more than 100 trypanosomes in the peritoneal cavity, only three persistent swimmers could be identified (*Figure 7B* and *Video 6*, 00:42 s - 01:35 min). Furthermore, trypanosomes were seen anchored by their posterior end to cells of the peritoneal membrane in a manner similar to that observed in the caudal vein (*Video 6*, 34 - 42 sec).

Next, we analysed the swimming behaviour of *T. carassii* in tissues: the tail tip, fins, muscle and interstitial space lining the blood vessels. Here, we observed no apparent consistency in swimming behaviour, and trypanosomes alternated between directional and non-directional swimming depending on the compactness of the tissue. For example, in the compact tissue of the

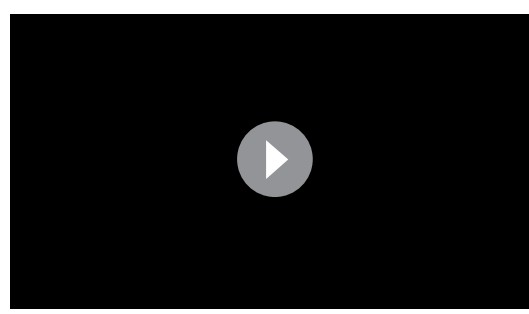

**Video 7.** Swimming behaviour of *T. carassii* in the narrow spaces of the fins, capturing persistent forward and backward swimmers.
DOI: https://doi.org/10.7554/eLife.48388.020

fins, most trypanosomes were directional swimmers (*Figure 8A–B*), although their path can often be interrupted or hindered by the compactness of the tissue. Swimmers moved at an average speed of 47.5 μm/s, covering up to 187 μm, before disappearing from view or colliding with an obstacle that resulted in a change in direction (*Video 7*, 0 – 47 s). Similar to the observations made in vitro (*Video 2*), trypanosomes could invert their swimming direction by swimming backwards (*Figure 8A–B*, *Video 7*, 00:47 s - 01:07 min). In vivo, backward swimming was only observed in the fins.

In the interstitial space lining the cardinal blood vessels (artery and vein), trypanosomes swim directionally or tumble (*Video 8*, 0 - 23 s), but can also use the space between cells to pin themselves and effectively invert swimming direction in a 'whip-like' motion, a movement distinct from the more random tumbling movement. (*Video 8*, 00:23 s - 01:17 min). Such 'whip-like' movement was also observed for trypanosomes swimming in ISC in which the blood flow is absent, and in more compact tissues such as the fin (*Figure 8C*, *Video 8*, 01:17 - 02:18 min). The 'whip-like' motion combines the swing of the flagellum along one plane, similar to the movement of tumblers on a glass

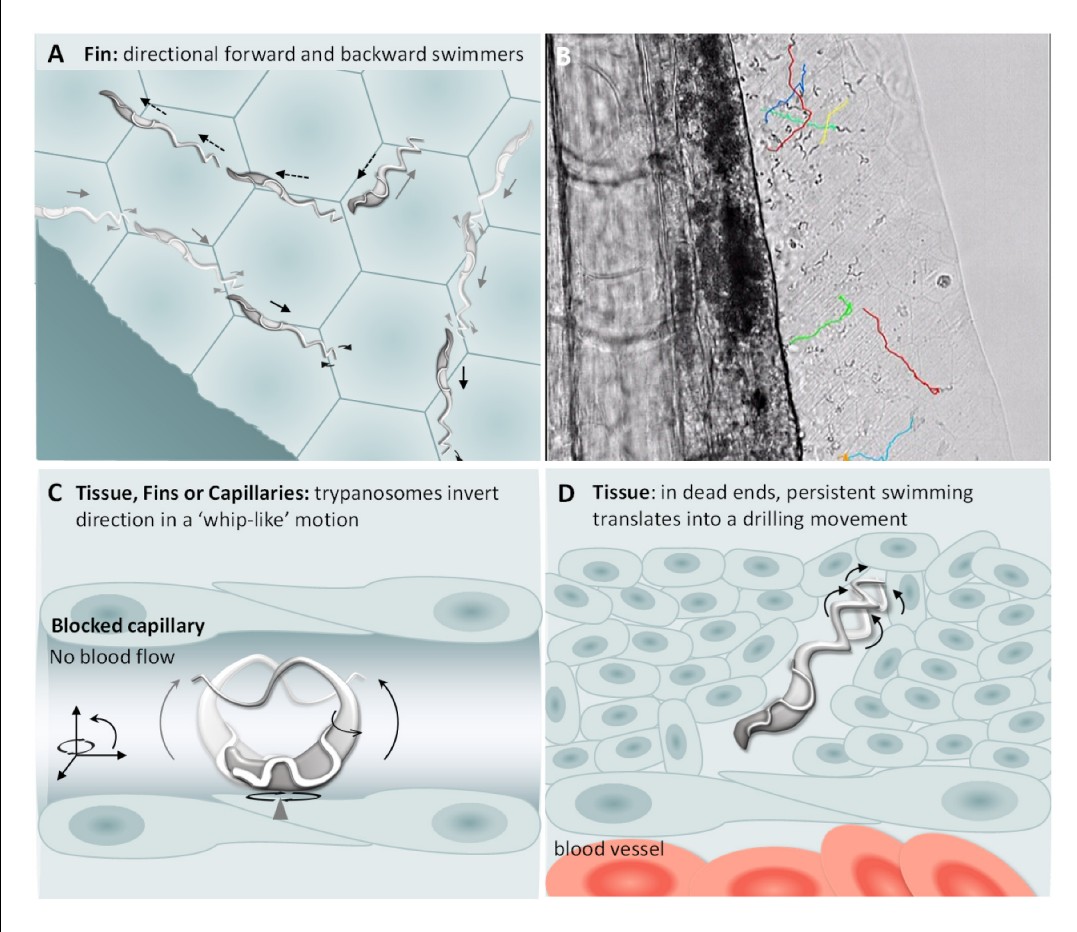

**Figure 8.** *T. carassii* swimming behaviour in tissues. (**A**) Schematic representation of *T. carassii* swimming in compact tissues such as those in the fins. Most trypanosomes are directional swimmers and both forward and backward swimming were observed. (**B**) Selected frame from *Video 7* showing the tracks of representative persistent swimmers identified in the fins. (**C**) In less compact tissues and in capillaries without blood flow, trypanosomes could invert their swimming direction in a 'whip-like' motion using the available three-dimensional space of the capillary or tissue. The 'whip-like' motion combines the swing of the flagellum along one plane (thin arc arrows), accompanied by a 180°C rotation of the flexible cell body along a third axis (rotational arrows). (**D**) In tissues where trypanosomes reach dead ends such as the interstitial space between vessels, persistent forward swimming translates into a drilling (auger) movement. *Video 7* and *Video 8* contain high-speed videos showing all locations and swimming behaviours schematically depicted.
DOI: https://doi.org/10.7554/eLife.48388.019

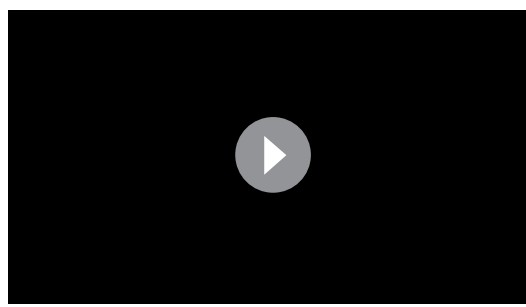

**Video 8.** Swimming behaviour of *T. carassii* in various tissues.
DOI: https://doi.org/10.7554/eLife.48388.021

surface (*Video 1*), accompanied by a 180°C rotation of the cell body along a third axis. This is indeed possible only in vivo where the cylindrical form of a capillary or interstitial space within a tissue allow the very flexible trypanosome cell body to move in three dimensions. Furthermore, in compact tissues that do not present a ready passage for trypanosomes, the persistent swimming translates into a drilling (auger) movement, which in some cases can lead to an enlargement of the space between somatic cells (*Figure 8D*, *Video 8*, 02:18 - 03:52 min).

Altogether, in tissues and tissue fluids *T. carassii* can adopt all swimming movements and can adhere through the posterior end to endothelial cells of the peritoneal cavity. Besides the previously described tumbling and directional (forward) swimming, trypanosomes were also able to invert direction through a 'whip-like' motion or by backward swimming.

## Progression of *T. carassii* infection and associated clinical signs

Physiological changes associated with the progression of the infection can affect the conditions within blood vessels or host tissues, and thus influence trypanosome behaviour. In addition to extravasation (*Videos 6–8*), which occurred as early as 1 dpi, we observed onset of anaemia and vasodilation of blood vessels.

Non-infected larvae have a strong and steady blood flow, with all blood vessels packed with red blood cells (*Figure 9A*). In contrast, infected larvae progressively showed a decrease in the ratio between red blood cells and trypanosomes from 1 to 8 dpi (*Figure 9B–C*), until in some cases red blood cells disappeared completely (*Figure 9D* and *Video 9*). Anaemia, therefore, is a hallmark of late stages of *T. carassii* infection in zebrafish larvae.

Highly infected fish that are anaemic also showed vasodilation, a clinical sign typical of advanced stages of infection (>3 dpi) with *T. carassii* (*Figure 10A–C*), most obviously observed in the caudal vein. The degree of vasodilation differed between individuals, and in extreme cases, the diameter of the caudal vein could be up to three times larger than that of control fish (*Figure 10D*). Vasodilation also occurs in the caudal artery but to a lesser extent (not shown). Interestingly, while the dilated blood vessels of larvae are packed with trypanosomes and have limited circulation, the number of extravasated trypanosomes is very low (*Video 9*).

## Discussion

In this study we describe a trypanosome infection model in zebrafish. By combining the transparency of zebrafish larvae with high spatio-temporal resolution microscopy, we were able to describe in detail the in vivo swimming behaviour of *T. carassii* in blood, tissues and tissue fluids of a vertebrate host. In addition to non-directional tumbling and directional forward swimming, we also describe how in vivo trypanosomes can reverse direction through a 'whip-like' motion or by swimming backwards. Finally, we report a novel observation that the posterior end of *T. carassii*, possibly coinciding with the flagellum base or flagellar pocket's neck, can act as an anchoring site. To our knowledge, this is the first report of the swimming behaviour of trypanosomes in vivo in a vertebrate host environment.

Knowledge of trypanosome-host cell interaction, movement, and tropism in vertebrate hosts

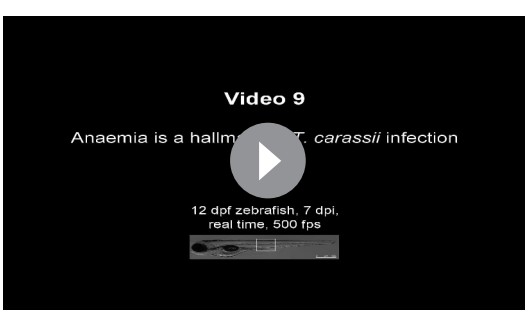

**Video 9.** Anaemia is a hallmark of *T. carassii* infection.
DOI: https://doi.org/10.7554/eLife.48388.023

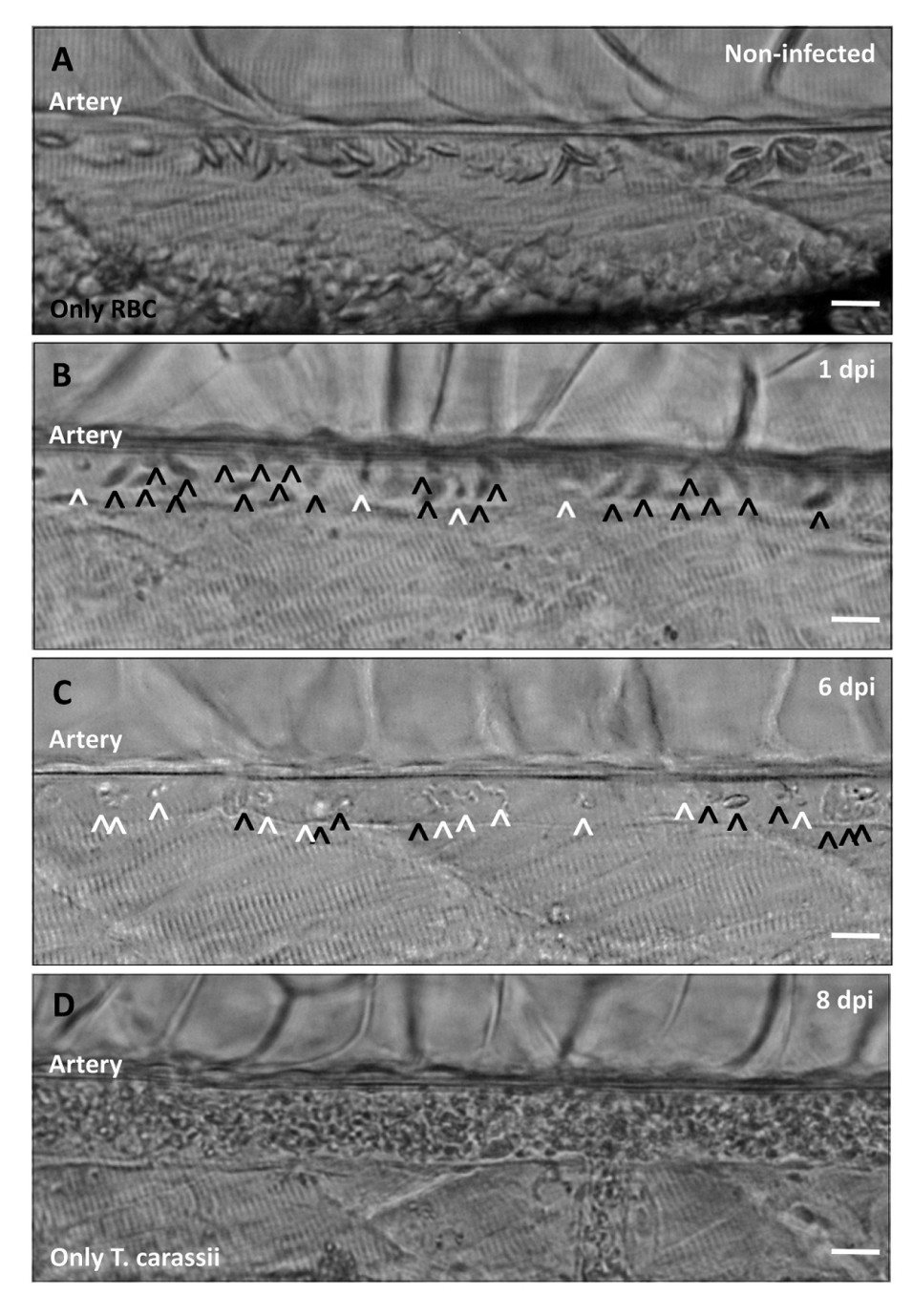

**Figure 9.** Onset of anaemia during *T. carassii* infection. Zebrafish larvae (5 dpf) were infected with 200 *T. carassii* or injected with PVP as non-infected control. Images are selected frames depicting the caudal artery, extracted from high-speed videos where trypanosomes (white open-arrow heads) and red blood cells (RBC, black open-arrow heads) were identified and tracked. (A) Artery of a control, non-infected, fish. Only RBC are present. (B) Artery of an infected fish, 1 dpi, showing a high ratio of RBC:trypanosomes. This frame corresponds to seconds 04:20-04:23 in *Video 3*, where the same trypanosomes were tracked. (C) Artery of an infected fish, 6 dpi, showing a reduced ratio of RBC:trypanosomes, indicating the onset of anaemia. (D) Artery of an infected fish suffering from severe anaemia, 8dpi, where only trypanosomes are present. The frame is extracted from the corresponding *Video 9*. Scale bars indicate 25 μm.

DOI: https://doi.org/10.7554/eLife.48388.022

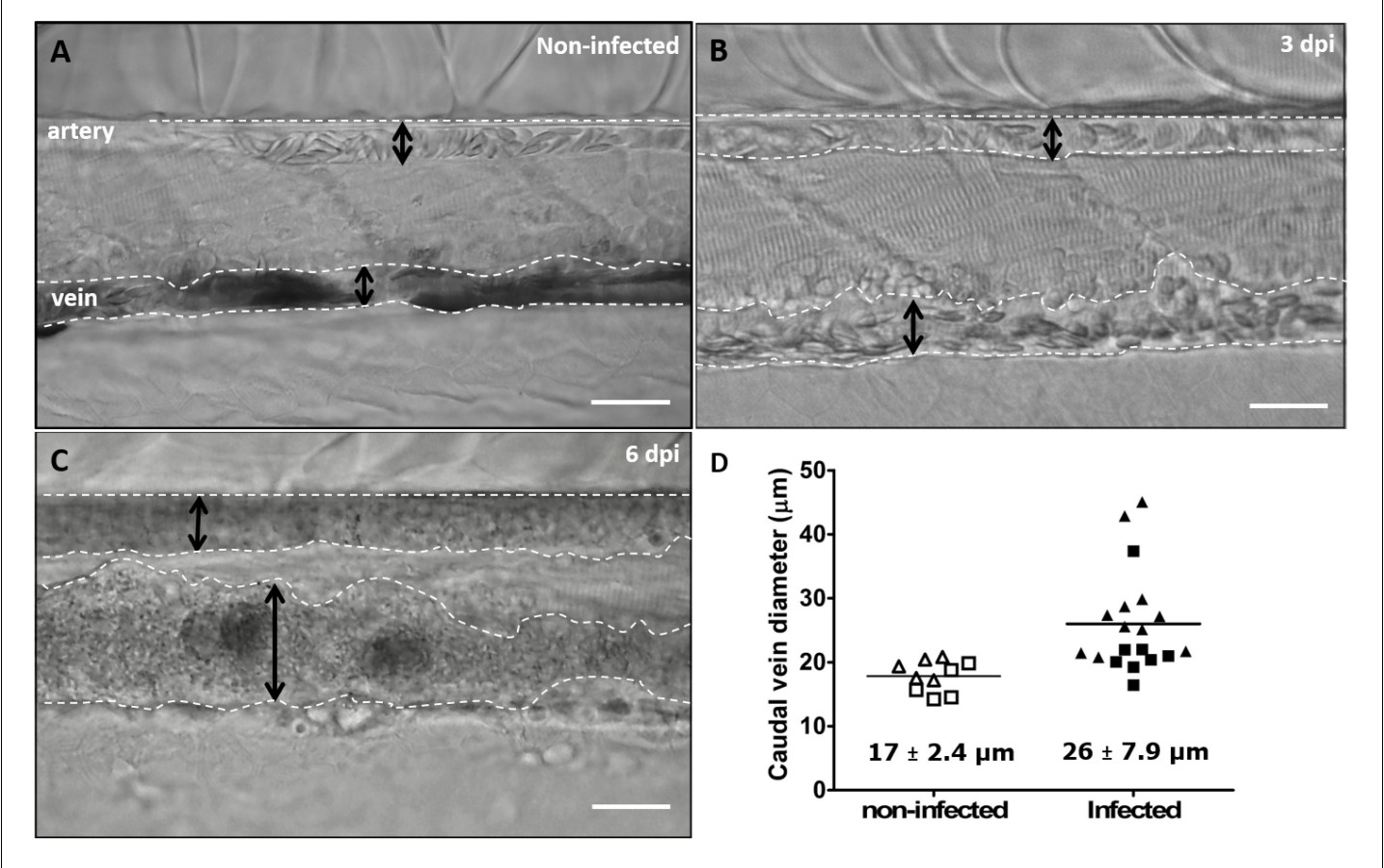

**Figure 10.** Advanced stages of *T. carassii* infection lead to vasodilation of the caudal vein. Wild type zebrafish larvae (5 dpf) were infected with 200 *T. carassii* or injected with PVP as non-infected control. Images are selected frames from high-speed videos. (**A-C**) Representative images of caudal artery and caudal vein (dashed lines) region at various time points after infection. Scale bars indicate 50 µm. (**D**) Maximum diameter of the caudal vein in non-infected (open symbols) and infected individuals (closed symbols) at 2–3 dpi (squares) and 6–8 dpi (triangles). Each value is the average of at least three measurements taken at different locations within the caudal vein of the same individual. Numbers indicate average and standard deviation.
DOI: https://doi.org/10.7554/eLife.48388.024

The following source data is available for figure 10:

**Source data 1.** Vasodilation in T. carassii infected larvae.
DOI: https://doi.org/10.7554/eLife.48388.025

is important to understand trypanosome biology and pathology. To date, detailed analysis of trypanosome swimming behaviour and interaction with vertebrate host cells has only been possible using isolated blood or conditions that best mimic those of the host blood or tissues. Although the current in vitro approaches have brought a wealth of information on the quantitative aspects of trypanosome motility and potential immune evasion strategies, they could not fully reproduce the streaming nature of the blood, the heterogeneity of the bloodstream between and within vessels, the different sizes of blood vessels between which trypanosomes regularly alternate, the different types of endothelium lining arteries and veins and, finally, the changes that occur in the blood caused by the infection itself.

In this study, we first investigated the motion of *T. carassii* in infected carp blood. Based on previously proposed descriptions of the swimming behaviour of salivarian trypanosomes in mouse blood (*Bargul et al., 2016*; *Shimogawa et al., 2018*), clearly more than 90% of *T. carassii* could be classified as tumblers, and this was independent of whether they were in whole blood, diluted in serum or culture medium. Our observations are in agreement with a previous study reporting that directional persistent swimming was not a prominent feature of *T. brucei* in whole blood from an

immunocompetent infected mouse (*Shimogawa et al., 2018*), but are in contrast to the study by Bargul and colleagues in which up to 30% persistent swimmers and 45% intermediate swimmers could be observed in blood films from immunosuppressed infected mice (*Bargul et al., 2016*). Differences in *T. brucei* strains, host immune status as well as the use of whole blood or blood films may account for the observed discrepancies. Nevertheless, both studies could not reproduce the streaming of the blood and, thus, the observations were made under static conditions.

Our initial in vitro observations led to the suggestion that the tumbling behaviour might be an intrinsic property of *T. carassii* in the bloodstream. However, our subsequent in vivo observations showed that this applies only to trypanosomes swimming in fluids with highly reduced or absent flow, for example the peritoneal fluid or blood in vessels in which the flow was slow or absent (*Figure 7A* and *Video 9*). In contrast, analysis of the swimming behaviour of *T. carassii* in the zebrafish bloodstream revealed that it is not possible to assign a single or predominant swimming behaviour. In vessels with a normal blood flow and in the presence of red blood cells, trypanosomes were dragged passively by the flow along with red blood cells. In the centre of large vessels, such as the cardinal artery or cardinal vein, the velocity of the flow does not allow either tumbling or directional swimming against or faster than the flow. Similarly, in narrow iISCs, trypanosomes are forced to move forward because of streaming of the blood or the presence of colliding red blood cells. Under these conditions, trypanosomes were never seen swimming faster than the passive movement of red blood cells as previously suggested (*Heddergott et al., 2012*; *Langousis and Hill, 2014*). Furthermore, because trypanosomes are carried along with red blood cells by the flow, it is difficult to envisage how the red blood cells could represent anything more than very occasional obstacles or surfaces for mechanical interactions that would favour forward swimming (*Bargul et al., 2016*; *Heddergott et al., 2012*). At the periphery of the cardinal vein, there is a cell-free layer where the number of red blood cells and the speed of the blood flow is reduced (*Bagchi, 2007*). Here, trypanosomes could clearly slow themselves down by crawling, rolling or by temporarily anchoring themselves to the dorsal endothelium (further discussed below). However, in arteries or small capillaries, adherence to the epithelium was never observed, even when the flow was reduced or completely absent. This suggests that not only the velocity of the flow but also the type of endothelium influences trypanosome swimming behaviour.

Physiological changes occur as the infection progresses that modify conditions within blood vessels. The flow is strongly reduced or interrupted and the density of red blood cells is reduced by obstruction of blood vessels or anaemia. In these conditions, directional swimmers were observed that swam faster than the flow and others that swam against the flow, or that repeatedly changed direction within blood vessels (*Video 4*). Taken together, our observations show that it is not possible to generalize the swimming behaviour of trypanosomes in blood as they can rapidly adopt different swimming behaviours: tumbling, directional swimming or anchoring. These behaviours are all largely influenced by the size and type of vessel, the speed of the flow, the presence, or not, of red blood cells as well as the highly dynamic microenvironment within a vessel, for example the centre compared to the periphery, and the type of endothelium.

Outside of the blood, it has been suggested that the directionality of swimming is influenced by the density or compactness of the tissue (*Bargul et al., 2016*; *Sun et al., 2018*; *Wheeler et al., 2013*). Our observation of swimming behaviour in various tissues revealed that it is the size of the interstitial space through which the trypanosome has to swim that largely determines whether it will swim directionally, tumble or both. For example, within the compact tissue of the fins, where the space below epithelial cells is very narrow (mesenchyme) and the basement membrane is not deformable (*Mateus et al., 2012*), the vast majority of trypanosomes swim directionally. This observation is in agreement with the swimming behaviour described for procyclic forms of *T. brucei* swimming through microfluidic devices smaller than their maximum cell diameter, mimicking potential size-limiting environments within host tissues (*Sun et al., 2018*). In zebrafish fins, the space can be so restrictive that to invert direction the trypanosome is obliged to swim backwards (*Video 7*). Backward swimming was previously observed in vitro or ex vivo for *T. brucei* motility mutants or for wild type trypanosomes swimming in high-density medium or in whole blood (*Bargul et al., 2016*; *Baron et al., 2007*; *Branche et al., 2006*; *Engstler et al., 2007*; *Heddergott et al., 2012*; *Shimogawa et al., 2018*), and ex vivo for procyclic and mesocyclic stages of *T. brucei* parasites swimming in confined spaces within the midgut of the tsetse fly (*Schuster et al., 2017*). The

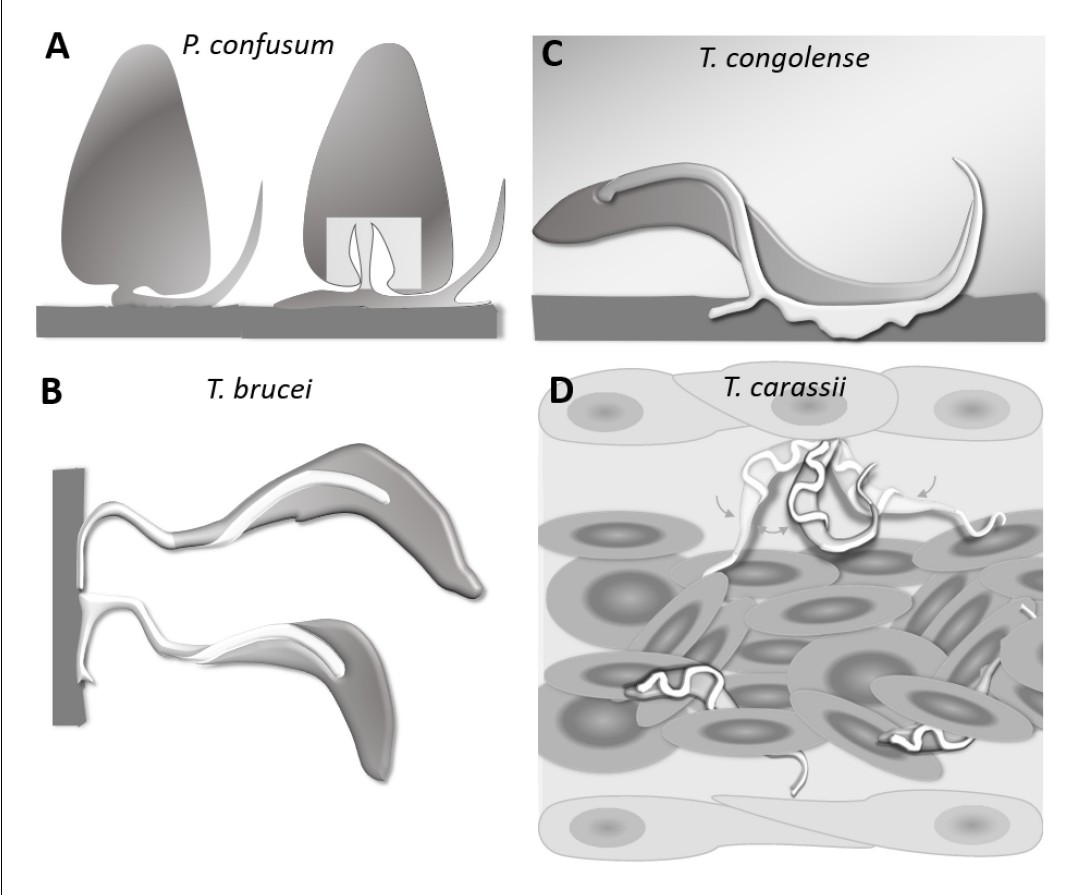

**Figure 11.** Schematic drawing depicting the attachment of various trypanosome species. (A) Haptomonads stages of *Paratrypanosoma confusum*: adhesion occurs through an attachment pad forming from the bulge at the base of the flagellum involving extensive remodelling of the flagellum membrane (based on *Skalický et al., 2017*). The square indicates the location of the flagellar pocket. (B) *T. brucei* epimastigotes attached through the flagellum to the brush border of the salivary gland epithelium (based on *Beattie and Gull, 1997*; *Schuster et al., 2017*; *Vickerman and Tetley, 1990*). (C) *T. congolense* adhesion to bovine aorta endothelial cell line via extensive membrane protrusions (filopodia) of the membrane-attached flagellum (based on *Beattie and Gull, 1997*; *Hemphill and Ross, 1995*). (D) *T. carassii* attached through the posterior end, leaving the cell body and flagellum free to move, as also shown in *Figure 5*.
DOI: https://doi.org/10.7554/eLife.48388.026

observation of *T. carassii* swimming in zebrafish tissues is the first report of backward swimming in a vertebrate host environment.

In the less compact areas of the fins or in the tissue lining the blood vessels, in addition to directional swimmers (forward or backwards), tumblers and trypanosomes that repeatedly inverted direction through a whip-like motion were observed (*Video 8*). When directional swimmers reached dead ends, the persistent forward swimming translated into either a drilling movement similar to the movement described for *T. brucei* in dead-end spaces within the midgut of the tsetse fly (*Schuster et al., 2017*), or backwards swimming as observed in the fins. Altogether, these observations indicate that trypanosomes can adopt several swimming behaviours and that these are largely influenced by the compactness and confinement offered by the tissue.

Perhaps one of the most interesting observations is the discovery that trypanosomes can anchor themselves to the vein endothelium by their posterior end, leaving the flagellum and the entire cell body free to move. Anchoring was observed as soon as 30 min to 1 h after *T. carassii* injection into zebrafish larvae, and by 1 dpi extravasation was observed. Under the current conditions, it was not possible to ascertain whether this adhesion mechanism favours or is even required for extravasation. So far, we have been unable to capture the exact moment of extravasation.

The anchoring site on the posterior of the trypanosome leaves the cell body and flagellum free to move (*Video 2* and *Video 5*), so it seems likely that the adhesion on the trypanosome occurs via the flagellum base or the neck of the flagellar pocket itself. The presence of specific adhesion molecules would at least partly explain how *T. carassii* can very rapidly anchor themselves, upon sudden collision with the endothelium, and remain in position for more than 15 s despite the very rapid blood flow and collisions with blood cells. Anchoring was only observed in the cardinal vein but trypanosomes were seen crawling on vein endothelium (*Video 5*) through dynamic interactions that involved the cell membrane, not just the flagellum membrane, suggesting that the molecules required for whole-cell adherence might be different from those required for anchoring through the posterior end. This is reminiscent of leukocyte rolling, which also occurs only in veins and not in arteries. Altogether, the possibility to observe *T. carassii* behaviour both in vitro and in vivo, in the presence or absence of a strong hydrodynamic flow, and at various locations within the vertebrate host, demonstrated how the environmental conditions, especially the presence or absence of a flow, strongly influence the ability of the trypanosome to attach and the duration of the attachment.

*T. carassii* anchors to zebrafish cells in a manner clearly distinct from that described for other trypanosomes. The major difference being the lack of extensive interaction between the trypanosome's flagellum membrane and the host cell or artificial surface. Stable interaction involving large portions of the flagellum membrane has been described among others for haptomonads stages (surface-attached) of *Paratrypanosoma confusum*, or *Leishmania* promastigotes (*Figure 11A*) (*Skalický et al., 2017*; *Wakid and Bates, 2004*). In these liberform parasites (flagellum not laterally attached to the cell), adhesion occurs through an attachment pad forming from the bulge at the base of the flagellum. At least in vitro, the formation of the pad takes approximately 1 h, causing extensive remodelling of the flagellum itself, and effectively anchors the parasites to the surface, and in the case of *P. confusum*, also favours their division (*Skalický et al., 2017*). Similarly, *T. brucei* epimastigotes divide while attached to the brush border of the salivary gland epithelium through extensive outgrowths of the non-cell-attached anterior part of the flagellum membrane (*Figure 11B*) (*Beattie and Gull, 1997*; *Langousis and Hill, 2014*; *Schuster et al., 2017*). Such an adhesion mechanism was only recently captured ex vivo through high-speed videography of dissected tsetse fly salivary glands; however, the exact moment of attachment and the time required to establish the stable interaction in vivo were not reported (*Schuster et al., 2017*). *T. congolense* was reported to adhere in vitro to bovine aorta endothelial (BAE) cell monolayers via extensive membrane protrusions (filopodia) of the membrane-attached flagellum (*Figure 11C*) (*Beattie and Gull, 1997*; *Hemphill and Ross, 1995*), an interaction that was shown to involve sialic acid residues on BAE cells (*Hemphill et al., 1994*). Although adhesion was observed already at 1 h, the filopodia increased in size over a period of 24–48 h. Whether such a type of interaction occurs with similar kinetics also in vivo, in vessels with an intact blood flow, is yet to be confirmed.

All the above attachment mechanisms are clearly geared towards creating a very stable interaction with a surface to either establish a permanent infection in the salivary glands of the insect host, or to possibly adhere to the vertebrate host endothelium. They all involve extensive modifications of the flagellum membrane that occur over time to increase the contact area between the (para)trypanosomes and the surface. Most of these interactions, however, were observed in vitro for (para)trypanosomes cultivated on glass or plastic surfaces or endothelial cell monolayers and studied by means of scanning or transmission electron microscopy, as well as in vitro binding assays (*Beattie and Gull, 1997*; *Hemphill and Ross, 1995*; *Hemphill et al., 1994*; *Skalický et al., 2017*; *Vickerman, 1969*; *Wakid and Bates, 2004*). Therefore, to what extent the kinetics of interaction and the size of the contact area described also apply to the more dynamic in vivo environment is yet to be ascertained. The interactions between *T. carassii* and zebrafish cells were limited to the tip of the posterior end of the trypanosome body (*Figure 11D*), leaving the entire flagellum and trypanosome cell body free to rapidly move (*Video 2*). Despite the small surface involved, anchoring occurred very rapidly, suggesting a very strong, yet dynamic, type of interaction, the duration of which was influenced by the presence or absence of a strong hydrodynamic flow as well as colliding red blood cells. Given the in vivo dynamic conditions within a blood vessel, it seems unlikely that *T. carassii* would establish stable interactions that involve large portions of the trypanosome cell surface, as for example described in vitro for *T. congolense*. We were unable to determine whether the anchoring area of *T. carassii* corresponds to the flagellum base, flagellar pocket neck or to the cell body membrane. Nevertheless, because *T. carassii* adherence to red blood cells and to glass

surfaces was also observed in vitro, it will be possible to investigate the adhesion mechanism at high resolution in the future.

In conclusion, we describe here for the first time the swimming behaviour of a trypanosome in vivo in the natural environment of a vertebrate host. We report the complex and heterogeneous environment in which the trypanosomes reside and how this highly influences the swimming behaviour. We describe how it is not possible to assign specific behaviours to trypanosomes swimming in any of the host compartments, as the trypanosomes were extremely effective in rapidly adapting their motion to the highly dynamic host environment. We describe backward swimming and whip-like movements that allowed trypanosomes to invert direction as well as identifiedying the posterior end as a novel anchoring site that allows the cell to adhere to host cells in a manner different from those described to date for other trypanosomes. Altogether, establishment of the *T. carassii* zebrafish infection model in combination with the genetic tractability of the zebrafish and of trypanosomes, represent a unique possibility to address questions related to: 1) trypanosome swimming behaviour in vivo in the natural environment of a vertebrate host, 2) host-pathogen interaction, 3) trypanosome biology, 4) the effect of specific immune factors on the progression of the infection, 5) the effect of drugs on both the trypanosome and the host, and 6) immune evasion strategies of trypanosomes that do not present antigenic variation. For all these reasons, the *T. carassii*-zebrafish model holds the promise to become a valuable complementary model to those currently available, to study the complex biology of trypanosomes and their interaction with the vertebrate host.

## Acknowledgements

The authors wish to thank the CARUS-ARF Aquatic Research Facility of Wageningen University for fish rearing and husbandry.

## Additional information

### Funding

| Funder | Grant reference number | Author |
|---|---|---|
| H2020 Marie Skłodowska-Curie Actions | PITN-GA-2011-289209 | Éva Dóró<br>Geert F Wiegertjes<br>Maria Forlenza |
| Nederlandse Organisatie voor Wetenschappelijk Onderzoek | 022.004.005 | Sem H Jacobs |

The funders had no role in study design, data collection and interpretation, or the decision to submit the work for publication.

### Author contributions

Éva Dóró, Maria Forlenza, Conceptualization, Data curation, Formal analysis, Supervision, Funding acquisition, Methodology, Writing—original draft, Project administration, Writing—review and editing; Sem H Jacobs, Conceptualization, Data curation, Formal analysis, Methodology, Writing—review and editing; Ffion R Hammond, Conceptualization, Data curation, Formal analysis, Methodology, Writing—original draft, Writing—review and editing; Henk Schipper, Data curation, Software, Formal analysis, Methodology, Writing—review and editing; Remco PM Pieters, Conceptualization, Data curation, Software, Formal analysis, Methodology, Writing—review and editing; Mark Carrington, Conceptualization, Software, Methodology, Writing—review and editing; Geert F Wiegertjes, Conceptualization, Software, Supervision, Funding acquisition, Methodology, Writing—original draft, Project administration, Writing—review and editing

### Author ORCIDs

Sem H Jacobs http://orcid.org/0000-0001-7482-3438
Mark Carrington http://orcid.org/0000-0002-6435-7266
Maria Forlenza https://orcid.org/0000-0001-9026-7320

## Ethics

Animal experimentation: All animals were handled in accordance with good animal practice as defined by the European Union guidelines for handling of laboratory animals (http://ec.europa.eu/ environment/ chemicals/lab_animals/home_en.htm). All animal work at Wageningen University was approved by the local experimental animal committee (DEC number 2014095).

## Decision letter and Author response

Decision letter https://doi.org/10.7554/eLife.48388.029
Author response https://doi.org/10.7554/eLife.48388.030

## Additional files

### Supplementary files

• Transparent reporting form
DOI: https://doi.org/10.7554/eLife.48388.027

### Data availability

All data generated or analysed during this study are included in the manuscript. Source data files have been provided for Figures 1, 2 and 8.

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
