## [Decision Letter]

Thank you for submitting your article "Visualising trypanosomes in a vertebrate host reveals novel swimming behaviours, adaptations and attachment mechanisms" for consideration by *eLife*. Your article has been reviewed by three peer reviewers, and the evaluation has been overseen by a Reviewing Editor and Anna Akhmanova as the Senior Editor. The reviewers have opted to remain anonymous.

The reviewers have discussed the reviews with one another and the Reviewing Editor has drafted this decision to help you prepare a revised submission.

Summary:

The reviewers agreed that this is a very interesting paper that for the first time describes trypanosome movements within a vertebrate host. One reviewer said "It addresses an important problem, is innovative, rigorous, and produces important and foundational results. The text is well-written and results mostly well-supported". Another said "The word "descriptive" is often used in a pejorative sense by reviewers to kill a manuscript. This manuscript is almost purely descriptive; nevertheless it serves as a timely reminder that precise observation and description can be of great value."

Essential revisions:

– Introduction section: The authors cite several papers as having studied trypanosomes in blood from an infected animal. Many of these however, use blood, but not from infected animals. Text could be corrected by saying blood "…from naive or infected animal…". Also, a recent paper (Shimogawa et al., 2018 – cited elsewhere in the manuscript) should be included here, as this work does study parasite movement in blood from an infected animal.

– Subsection “Characterization of trypanosome swimming behaviour in vitro” –, regarding classification of "swimming behavior". The term "tumbling swimmers" does not fit the data/description, because "non-directional movement" doesn't involve 'swimming'. Also, many video captions use the term "tumbler" (see e.g. Video 1, title). Finally, based on the data and descriptions provided (e.g. multiple videos, and elsewhere), it seems that the cells are either swimming (directional movement), tumbling (non-directional movement), or alternating between these two states. Therefore, I would recommend "persistent swimmers"; "persistent tumblers" (or simply "tumblers"); and "intermediate swimmers"

– Cell morphology is not an appropriate aspect to be used in assigning swimming classification. Two reasons: 1) the classification is about swimming, not morphology; 2) Tumblers can be seen to have a "stretched-out" cell body form, e.g. one can see frames within Video 1 of "tumbling" cells showing a stretched out form. (bottom right cell in 'tumbler' section at 0:00:20 of Video 1; middle/left cell in 'tumbler' section at 0:00:25 of Video 1).

– This is a bit of a nuance, but please also distinguish shape (e.g. circle or square) vs form/morphology (e.g. sphere or cube), in description of cell appearance.

– Figure 1 and various places in text: I don't think it is accurate or appropriate to ascribe a "speed" to tumbling cells. In Video 1 for example, the tumblers shown may exhibit displacement of one point of the cell over time, but ascribing 'speed' to this is not analogous to 'speed' ascribed to movement of the cell as a whole in persistent swimmers.

– Discussion second paragraph: should keep this to “vertebrate” host, because description in vivo with tsetse fly cells has been described.

– Discussion third paragraph: Recent work, Shimogawa et al., 2018, should be cited also, as that is distinguished by using infected blood from immunocompetent mice, and thus is distinguished from the also important Bargul work. Furthermore, the shimogawa studies find that, as in the current study, persistent swimmers are relatively rare.

– Given the extended attention given to the observation that parasite movement is impacted by movement on tissue surfaces, e.g. fin and tail, some reference to studies examining parasite motility on surfaces should be provided.

– Subsection “Characterization of trypanosome swimming behaviour in vitro”: The statement that trypanosomes move backward "without using the propelling motion of the flagellum." is not supported. In the video referred to, the backward movement is almost certainly due to the propelling motion of the flagellum, i.e. a base-to-tip beat that drives cell movement with the base of the flagellum leading. A similar 'backward' motility has been described on numerous occasions for T. brucei.

– "whip-like" movement. This is described as something novel but appears to be very similar to rapid flagellum tip movements described previously, e.g. Rodriguez et al., 2009. The impression that this is a novel movement, in the first paragraph of the Discussion, should omitted.

– Backward motility. The implication that backward motility is something not describe previously, e.g. in the first paragraph of the Discussion, should be omitted. Likewise, related in paragraph four, please note that backward motility has been reported not just for mutants, but also for WT cells ex vivo, e.g. Bargul et al., 2012; Shimogawa et al., 2018.

– One reviewer requested that the authors to put some gallery of images of selected time frames in addition to the videos. This could ideally guide the reader's eye while reading.

---

## [Author Response]

Essential revisions:– Introduction section: The authors cite several papers as having studied trypanosomes in blood from an infected animal. Many of these however, use blood, but not from infected animals. Text could be corrected by saying blood "…from naive or infected animal…". Also, a recent paper (Shimogawa et al., 2018 – cited elsewhere in the manuscript) should be included here, as this work does study parasite movement in blood from an infected animal.

Thank you, we have amended the text and inserted the reference as suggested.

– Subsection “Characterization of trypanosome swimming behaviour in vitro”, regarding classification of "swimming behavior". The term "tumbling swimmers" does not fit the data/description, because "non-directional movement" doesn't involve 'swimming'. Also, many video captions use the term "tumbler" (see e.g. Video 1, title). Finally, based on the data and descriptions provided (e.g. multiple videos, and elsewhere), it seems that the cells are either swimming (directional movement), tumbling (non-directional movement), or alternating between these two states. Therefore, I would recommend "persistent swimmers"; "persistent tumblers" (or simply "tumblers"); and "intermediate swimmers"

Thank you for the suggestions, we opted to refer to “tumblers” and removed from the main text and videos any reference to tumbling swimmers.

– Cell morphology is not an appropriate aspect to be used in assigning swimming classification. Two reasons: 1) the classification is about swimming, not morphology; 2) Tumblers can be seen to have a "stretched-out" cell body form, e.g. one can see frames within Video 1 of "tumbling" cells showing a stretched out form. (bottom right cell in 'tumbler' section at 0:00:20 of Video 1; middle/left cell in 'tumbler' section at 0:00:25 of Video 1).

We removed any reference to cell body shape in the section pertaining the description of the different swimming behaviours and from the corresponding Video 1. In other sections of the paper (related to Video 3 and Figure 3 for example), we describe that parasites ‘curl or stretch the cell body while being passively dragged by the flow’, but this is not linked to a specific swimming behaviour (tumbling or directional swimming). Thus, we kept those terms, as they best describe what we observe in the blood.

– This is a bit of a nuance, but please also distinguish shape (e.g. circle or square) vs form/morphology (e.g. sphere or cube), in description of cell appearance.

By addressing the previous point, we also incorporated the changes requested in the current one. Furthermore, we scanned the manuscript for similar mistakes elsewhere and amended them.

– Figure 1 and various places in text: I don't think it is accurate or appropriate to ascribe a "speed" to tumbling cells. In Video 1 for example, the tumblers shown may exhibit displacement of one point of the cell over time, but ascribing 'speed' to this is not analogous to 'speed' ascribed to movement of the cell as a whole in persistent swimmers.

We agree that indeed speed is not analogous for the two types of cell, nevertheless, we wanted to describe that tumblers are very mobile cells and looked for a way to quantify such motility. Speed of tumblers was reported previously in at least two papers (Heddergott et al., 2012; Bargul et al., 2016), and given the availability of high-speed videos that allowed us to accurately follow the displacement of the posterior end of *T. carassii* in space and time, we would like to keep the description of speed to reflect their mobility.

Nevertheless, to clarify to the reader the difference between the speed of tumblers and that of directional swimmers, we have revised the main text and the legend of Figure 1.

– Discussion second paragraph: should keep this to “vertebrate” host, because description in vivo with tsetse fly cells has been described.

The text has been revised.

– Discussion third paragraph: Recent work, Shimogawa et al., 2018, should be cited also, as that is distinguished by using infected blood from immunocompetent mice, and thus is distinguished from the also important Bargul work. Furthermore, the shimogawa studies find that, as in the current study, persistent swimmers are relatively rare.

Thank you for the suggestion, indeed we missed this important observation in the study. The reference has been added and further elaborated upon in the Discussion.

– Given the extended attention given to the observation that parasite movement is impacted by movement on tissue surfaces, e.g. fin and tail, some reference to studies examining parasite motility on surfaces should be provided.

We are not aware of any study specifically studying or visualizing trypanosome swimming behaviour on surfaces mimicking the conditions present in the tissue of its host. We refer however to the study by Schuster et al., 2017 studying swimming behaviour of *T. brucei* in various compartments of the Tsetse fly.

Two additional studies used micropillars and microfluidic assays to investigates how bloodstream forms (Bargul et al., 2016) or procyclic forms (Sun et al., 2018) of trypanosomes would swim between obstacles or through orifices smaller than their maximum diameter. Although the study by Bargul used pillars to simulate the distance between red blood cells, RBC are not static objects and thus the pillars system mimics better the situation in a tissue than in the blood. The study by Sun uses insect procyclic stages and indeed predicted very well the ability of trypanosomes to penetrate very compact tissues and to swim directionally.

Last but not least we also referred already to the in vitro study reporting adherence of *T. congolense* to bovine aorta endothelial cells (Hemphill at al., 1995).

All above mentioned studies are now better integrated in the Discussion section.

– Subsection “Characterization of trypanosome swimming behaviour in vitro”: The statement that trypanosomes move backward "without using the propelling motion of the flagellum." is not supported. In the video referred to, the backward movement is almost certainly due to the propelling motion of the flagellum, i.e. a base-to-tip beat that drives cell movement with the base of the flagellum leading. A similar 'backward' motility has been described on numerous occasions for T. brucei.

The sentence "without using the propelling motion of the flagellum" has been removed from Video 2 and the corresponding main body text.

– "whip-like" movement. This is described as something novel but appears to be very similar to rapid flagellum tip movements described previously, e.g. Rodriguez et al., 2009. The impression that this is a novel movement, in the first paragraph of the Discussion, should omitted.

Supplementary movie 6 in the study by Rodriguez describes how bloodstream forms exhibit frequent and rapid swings of the flagellar tip and emphasizes the flexibility of the cell body. In our opinion, such movement is very similar to that of tumblers captured in Video 1 in our study, where trypanosomes can be seen ‘swinging’ very rapidly the flagellum and frequently ‘making contact’ with neighbouring RBC, although we think such contact mostly occurs randomly.

The ‘whip-like’ movement we describe is distinct from the above mentioned one since the ‘whip-like’ movement is used by trypanosomes in vivo to pin themselves, turn in a 3D motion and invert direction. The 3D motion combines the swing of the flagellum along one plane (similar to the one observed for tumblers on a glass surface in our study or the Rodriguez study), accompanied by a 180° rotation of the cell body along a 3^rd^ axis. This is indeed possible only in vivo where the cylindrical form of a capillary or interstitial space within a tissue, allow the parasite to move in three dimensions. In fact, we never observed a ‘whip-like’ movement during our in vitro observations, possibly due to the 2-D constraints of the surface.

To further clarify this to the reader we have amended the text as well as modified Figure 8C (previously 7C).

– Backward motility. The implication that backward motility is something not describe previously, e.g. in the first paragraph of the Discussion, should be omitted. Likewise, related in paragraph four, please note that backward motility has been reported not just for mutants, but also for WT cells ex vivo, e.g. Bargul et al., 2012; Shimogawa et al., 2018.

We apologize for this confusion, we wanted to emphasize that backward swimming was not described previously in vivo in a vertebrate host. The studies describing backwards movement in vitro or ex vivo have been referred to and we clarified that our statement strictly refers to our in vivo observations.

One reviewer requested that the authors to put some gallery of images of selected time frames in addition to the videos. This could ideally guide the reader's eye while reading.

*–* We have followed up on this suggestion and included a completely new Figure 2 showing selected frames from Video 2 capturing parasites attached to red blood cells and glass surfaces; We have split Figure 6, now Figure 7 and 8, and added selected frames from Video 6 and Video 7 showing trypanosomes swimming in the peritoneal cavity and in fins.